# DIMENSION-REDUCED ADAPTIVE GRADIENT METHOD

## ABSTRACT

Adaptive gradient methods, such as Adam, have shown faster convergence speed than SGD across various kinds of network models. However, adaptive algorithms often suffer from inferior generalization performance than SGD. Though much effort via combining Adam and SGD have been invested to solve this issue, adaptive methods still fail to attain as good generalization as SGD. In this work, we proposed a Dimension-Reduced Adaptive Gradient Method (DRAG) to eliminate the generalization gap. DRAG makes an elegant combination of SGD and Adam by adopting a trust-region like framework. We observe that 1) Adam adjusts stepsizes for each gradient coordinate, and indeed decomposes the $n$-dimensional gradient into $n$ independent directions to search; 2) SGD uniformly scales gradient for all gradient coordinates and actually has only one descent direction to minimize. Accordingly, DRAG reduces the high degree of freedom of Adam and also improves the flexibility of SGD via optimizing the loss along $k$ $(\ll n)$ descent directions, *e.g.* the gradient direction and momentum direction used in this work. Then per iteration, DRAG finds the best stepsizes for $k$ descent directions by solving a trust-region subproblem whose computational overhead is negligible since the trust-region subproblem is low-dimensional, *e.g.* $k = 2$ in this work. DRAG is compatible with the common deep learning training pipeline without introducing extra hyper-parameters and with negligible extra computation. Moreover, we prove the convergence property of DRAG for non-convex stochastic problems that often occur in deep learning training. Experimental results on representative benchmarks testify the fast convergence speed and also superior generalization of DRAG.

## 1 INTRODUCTION

SGD (Robbins & Monro, 1951) and its variant with momentum (Sutskever et al., 2013) are used widely in training deep neural networks. They perform well empirically and have theoretical guarantee (Szegedy et al., 2015; He et al., 2016; Lee et al., 2016; Hardt et al., 2016). However, SGD suffers from two issues. It often has slow convergence speed since it adopts a single learning rate for all the gradient coordinates. Moreover, it is also hard to tune the single learning rate (Wilson et al., 2017), since not all gradient coordinates share the same optimization properties.

To resolve this problem, several adaptive gradient methods have been proposed to adopt different learning rate for different gradient coordinates. Typical examples of such methods include Adagrad (Duchi et al., 2011), RMSProp (Tieleman et al., 2012), and Adam (Kingma & Ba, 2014). Emprically, these methods have shown faster convergence speed and eased the burden of carefully tuning the learning rate in SGD across many kinds of networks. However, their generalization performance are often worse than SGD in many scenarios (Wilson et al., 2017; Zhou et al., 2020).

Some algorithms are proposed to combine the fast convergence speed of adaptive gradient methods and good generalization performance of SGD. Instances of this type of algorithms include SWATS (Keskar & Socher, 2017) which automatically switchs from Adam to SGD, ND-Adam (Zhang et al., 2017) which utilizes vector learning rate and normalization to control direction and stepsize, and AMSGrad (Reddi et al., 2018) which maintains a monotone increasing second moment. Unfortunately, these methods only slightly bridge the generalization gap between SGD and Adam, but does not attain as good generalization performance as SGD, needless to say the state-of-the-art performance on test set. Accordingly, these algorithms are rarely used to train deep networks in practice.

To combine the merits of Adam and SGD, i.e. fast convergence speed in Adam and excellent generalization in SGD, we proposed a Dimension-Reduced Adaptive Gradient Method (DRAG for short) which minimizes the loss from several descent directions to trade-off the whole space search in Adam and the minimization along a single gradient direction in SGD. For Adam, adjusting stepsizes for each gradient coordinate actually transforms the $n$-dimensional gradient into $n$ independent directions to optimize, in which each direction inherits one coordinate element from the gradient and sets the remaining coordinate positions as zeros. In contrast, SGD only uses a single learning rate for all gradient coordinates and minimizes the loss along one descent direction. Though the adaptive learning rate for each coordinate shows faster convergence speed than a single learning rate for all coordinates, as shown in many works (Wilson et al., 2017; Zhou et al., 2018), it also leads to the inferior generalization in Adam, since minimizing $n$ independent directions means searching the whole parameter space and could results in overfitting. So it is natural to trade-off the number of descent directions.

To this end, motivated by DRSOM (Zhang et al., 2022), we update the parameters along the gradient direction and momentum direction through a trust-region-like approach, which greatly reduces the high adaptivity of Adam while adding flexibility to SGD. At each iteration, DRAG searches for the optimal update along the gradient and the momentum which are widely used in accelerated algorithm (Polyak, 1964; Nesterov, 2003) by solving a two-dimensional trust-region subproblem to find the best stepsizes for these two directions. For the trust-region subproblem, we use a quadratic approximation to estimate the loss with the Hessian matrix estimated by the second moment in Adam which is a diagonal matrix and can greatly reduce the computational cost. Moreover, we heuristically design a simple and effective trust-region radius for the subproblem. Despite the delicate design of our algorithm, we also theoretically prove that on non-convex problems, our DRAG can converge and enjoys a stochastic gradient complexity of $\mathcal{O}(\epsilon^{-4})$ to find an $\epsilon$-approximate first-order stationary point. To summarize, our contributions are as follows:

- We proposed the DRAG algorithm to optimize the loss from several descent directions for balancing the whole space search in Adam and the optimization along a single gradient direction in SGD. Moreover, we formulate the optimum stepsize search for these descent directions into a low-dimensional trust region problem whose computational cost is negligible when compared with the vanilla cost in adaptive gradient algorithms.

- We theoretically prove that to find an $\epsilon$-approximate stationary point on non-convex stochastic problems, DRAG has the stochastic gradient complexity of $\mathcal{O}(\epsilon^{-4})$ which matches the lower bound $\Omega(\epsilon^{-4})$ in (Arjevani et al., 2022) under the same non-convex optimization setting.

- Experimental results show that on several representative benchmarks, our DRAG method can achieve faster convergence speed than SGD, and also state-of-the-art generalization performance.

## 2 RELATED WORK

Adaptive gradient methods, *e.g.* Adam (Kingma & Ba, 2014), Adagrad (Duchi et al., 2011), and RMSprop (Tieleman et al., 2012), adopt different stepsizes for different gradient coordinates so as to boost training process. Although Adam and its variants are used widely for training deep neural networks, their poor generalization performance makes SGD still dominant in some areas, such as training CNNs (He et al., 2016). Many researchers tried to improve the generalization capacity of Adam, including SWATS (Keskar & Socher, 2017) which conducts an automatic switch from Adam to SGD training strategy, ND-Adam (Zhang et al., 2017) which controls the stepsizes and update direction in a more precise way, AMSGrad (Reddi et al., 2019) which ensures monotone increasing second moment, and Padam (Chen et al., 2021) which introduces a partially adaptive parameter to control the adaptivity of stepsizes. Our solution to improve the generalization performance is to confine the update of parameters to a two-dimensional subspace of the parameter space. Specifically, we solve a trust-region subproblem to determine the optimal stepsizes along the gradient direction and momentum direction at each iteration.

The idea of utilizing gradient and momentum direction to update the variable traces back to Polyak's heavy ball method (Polyak, 1964)

$$\boldsymbol{x}_t = \boldsymbol{x}_{t-1} - \alpha_1 \nabla f(\boldsymbol{x}_{t-1}) + \alpha_2 \boldsymbol{d}_{t-1}$$

SGD (with momentum) (Sutskever et al., 2013) can also be written in this form after replacing the gradient direction $\nabla f(\boldsymbol{x}_{t-1})$ with stochastic gradient direction $\boldsymbol{g}_{t-1}$. Unlike SGD which adopts constant stepsizes, DRAG adopts the stepsizes as the solution of a two-dimensional quadratic model with a spherical constraint, it adaptively learns the optimal stepsize for each direction.

Trust-region method has been widely used in optimization, and some works tries to use it for machine learning (Martens et al., 2010; Dudar et al., 2017; Erway et al., 2020). The closest related work is the Dimension Reduced Second Order Method (DRSOM) proposed by Zhang et al. (2022). It uses a finite difference method to approximate the Hessian-vector product arisen in the trust-region subproblem and achieves higher convergence rate than first order methods. However, difference method is inaccurate in high-dimensional problem, and it introduces huge extra computational overhead. Our method DRAG, on the other hand, approximates the Hessian directly by the second moment of stochastic gradient directly. It reduces extra forward-backward loops to calculate the Hessian-vector product in DRSOM and captures curvature information more accurately. Moreover, DRAG adopts a heuristic trust-region radius, making the algorithm compatible with dominant deep learning training pipeline without introducing extra hyperparameters.

## 3 METHOD

Training neural networks can be seen as solving the following non-convex optimization problem

$$\min_{\boldsymbol{x} \in \mathbb{R}^n} f(\boldsymbol{x}), \tag{1}$$

where $f$ is the loss function and $\boldsymbol{x} \in \mathbb{R}^n$ is the variable. Among all optimizers, Adam (Kingma & Ba, 2014) is one of the most popular algorithm to solve problem (1). At each training iteration, Adam maintains an exponential moving average (EMA) of first and second moments of stochastic gradient $\boldsymbol{v}_t$ and $\boldsymbol{u}_t$ as

$$\boldsymbol{v}_t = \beta_1 \boldsymbol{v}_{t-1} + (1 - \beta_1)\boldsymbol{g}_{t-1}, \quad \boldsymbol{u}_t = \beta_2 \boldsymbol{u}_{t-1} + (1 - \beta_2)\boldsymbol{g}_{t-1}^2,$$

where $\beta_1, \beta_2 \in [0, 1]$ are constant and $\boldsymbol{g}_{t-1} \coloneqq \tilde{\nabla} f(\boldsymbol{x}_{t-1})$ is the stochastic gradient. It adaptively scales the learning rates for each gradient coordinate, and actually minimizes the loss function along $n$ descent directions

$$\boldsymbol{x}_t = \boldsymbol{x}_{t-1} - \eta \frac{\hat{\boldsymbol{v}}_t}{\sqrt{\hat{\boldsymbol{u}}_t} + \nu} = \boldsymbol{x}_{t-1} - \sum_{i=1}^{n} \frac{\eta}{\sqrt{\hat{\boldsymbol{u}}_{t,i}} + \nu}(\hat{\boldsymbol{v}}_{t,i}\boldsymbol{e}_i), \tag{2}$$

where $\hat{\boldsymbol{v}}_t, \hat{\boldsymbol{u}}_t$ are bias-corrected $\boldsymbol{v}_t, \boldsymbol{u}_t$, $\boldsymbol{e}_i$ is the standard basis vector with 1 for dimension $i$ and 0 for all other dimensions. Specifically, Adam adopts a stepsize of $\frac{\eta}{\sqrt{\hat{\boldsymbol{u}}_{t,i}} + \nu}$ for the $i$-th gradient descent direction $\hat{\boldsymbol{v}}_{t,i}\boldsymbol{e}_i$.

While adaptive stepsize boosts the convergence of Adam, it weakens the generalization performance due to noise and overfitting. In contrast, SGD generalizes well because it uses a single stepsize for all gradient coordinates and indeed optimizes the loss function only along the gradient direction. One interpretation for their different generalization performance is that Adam's update direction no longer falls into the subspace spanned by all stochastic gradients $\text{span}\{\boldsymbol{g}_0, \cdots, \boldsymbol{g}_t\}$ (Wilson et al., 2017; Zhang et al., 2017), while SGD do. Actually, Wilson et al. (2017) proved that on a binary classification problem, SGD converges to the max-margin solution because its update at each step is linear combination of stochastic gradients, while adaptive gradient methods converge to solutions that generalize poorly because adaptivity makes the algorithm susceptible to noises and therefore causes overfitting.

To overcome the issue just mentioned, our DRAG algorithm optimizes the loss function in (1) from the gradient direction and the momentum direction. It maintains flexibility in the update direction while inheriting the generalization capacity of SGD. At each step, it searches for the optimal stepsizes along these two directions by solving a two-dimensional trust-region subproblem. Therefore, from the optimization perspective, it conducts the optimal update within the two-dimensional subspace spanned by gradient direction and momentum direction. Moreover, while DRAG adopts the trust-region framework, it is compatible with the dominant deep learning training pipeline without introducing extra hyperparameters.

### 3.1 DETAILS OF THE ALGORITHM

Details of our algorithm are described in Algorithm 1. At each training epoch, DRAG first computes stochastic gradient $g_{t-1}$, and use it to update the first moment $v_t$ and second moment $u_t$ of stochastic gradient like Adam. Then, we introduce the bias-corrected second moment $\hat{u}_t$ to approximate the Hessian. In this way, DRAG constructs the trust-region subproblem in line 9 of Algorithm 1. While solving this trust-region subproblem in high-dimensional parameter space is computational expensive, DRAG solves it in the two-dimensional subspace spanned by bias-corrected first moment direction $\hat{v}_t$ and momentum direction $d_{t-1}$, making the computational overhead negligible. Here we intuitively set the trust-region radius as $\eta \|\hat{v}_t\|$, and the benefits of this setting is described in Section 3.2. After calculating the solution $\alpha_{1t}$ and $\alpha_{2t}$ of the subproblem, we get an optimal update $p = -\alpha_{1t}\hat{v}_t + \alpha_{2t}d_{t-1}$ in the two-dimensional subspace. Finally, we follow (Loshchilov & Hutter, 2018) and conduct a decoupled weight decay step. This is the overall framework of our DRAG.

---

**Algorithm 1** Dimension-Reduced Adaptive Gradient Method (DRAG)

---

1: **Input:** Total number of training epoch $m$, learning rate $\eta$, exponential moving average coefficients $\beta_1, \beta_2$, weight decay scale $\gamma$, margin coefficient $\nu$.
2: **Initialize:** Set $x_0$, $v_0 = 0$, $u_0 = 0$.
3: **for** $t = 1, \cdots, m$ **do**
4: $\quad$ Compute stochastic gradient $g_{t-1} = \tilde{\nabla}f(x_{t-1})$.
5: $\quad$ $v_t = \beta_1 v_{t-1} + (1 - \beta_1)g_{t-1}$, $\hat{v}_t = v_t/(1 - \beta_1^t)$
6: $\quad$ $u_t = \beta_2 u_{t-1} + (1 - \beta_2)g_{t-1}^2$, $\hat{u}_t = u_t/(1 - \beta_2^t)$
7: $\quad$ $H_t = \text{diag}(\sqrt{\hat{u}_t} + \nu)$
8: $\quad$ $d_{t-1} = x_{t-1} - x_{t-2}$ if $t \geq 2$ else $d_{t-1} = 0$.
9: $\quad$ $(\alpha_{1t}, \alpha_{2t}) = \text{argmin}_p\{\langle \hat{v}_t, p\rangle + \frac{1}{2}\langle p, H_t p\rangle \mid \|p\| \leq \eta\|\hat{v}_t\|, \ p = -\alpha_1\hat{v}_t + \alpha_2 d_{t-1}\}$.
10: $\quad$ $x_t = x_{t-1} - \alpha_{1t}\hat{v}_t + \alpha_{2t}d_{t-1}$
11: $\quad$ $x_t = x_t - \eta\gamma x_{t-1}$ (Conduct weight decay)
12: **end for**
13: **Output:** $x_1, \cdots, x_m$

---

The only extra computational overhead of DRAG compared with Adam is solving the two-dimensional trust-region subproblem in line 9 of Algorithm 1. The trust-region subproblem can be formally formulated as follows:

$$
\begin{aligned}
\min_{\alpha_1, \alpha_2} \quad & \langle \hat{v}_t, -\alpha_1\hat{v}_t + \alpha_2 d_{t-1}\rangle + \frac{1}{2}\langle -\alpha_1\hat{v}_t + \alpha_2 d_{t-1}, H_t(-\alpha_1\hat{v}_t + \alpha_2 d_{t-1})\rangle \\
= \quad & \begin{bmatrix} \alpha_1 & \alpha_2 \end{bmatrix} \begin{bmatrix} -\hat{v}_t^T \hat{v}_t \\ \hat{v}_t^T d_{t-1} \end{bmatrix} + \frac{1}{2}\begin{bmatrix} \alpha_1 & \alpha_2 \end{bmatrix} \begin{bmatrix} \hat{v}_t^T H_t \hat{v}_t & -\hat{v}_t^T H_t d_{t-1} \\ -\hat{v}_t^T H_t d_{t-1} & d_{t-1}^T H_t d_{t-1} \end{bmatrix} \begin{bmatrix} \alpha_1 \\ \alpha_2 \end{bmatrix} \\
\text{s.t.} \quad & \|-\alpha_1\hat{v}_t + \alpha_2 d_{t-1}\| \leq \eta\|\hat{v}_t\|,
\end{aligned}
$$

where $H_t = \text{diag}(\sqrt{\hat{u}_t} + \nu)$ as defined in Algorithm 1. This two-dimensional subproblem can be solved efficiently by using its global minimal condition. In Appendix A, we transform this subproblem into a standard trust-region subproblem, and then an $\epsilon$-global primal-dual solution satisfying KKT condition can be found in $\mathcal{O}(\log\log(\frac{1}{\epsilon}))$ time (Luenberger et al., 1984). See more details in Appendix A.

If we set $p$ in line 9 of Algorithm 1 as

$$
p = -\alpha_1\hat{v}_t + \sum_{j=1}^{k-1} \alpha_{j+1}d_{t-j},
$$

our algorithm DRAG can be generalized to solve the subproblem with any $k$ search directions. Although according to experimental results, one search direction or multiple search directions usually perform worse than two search directions (DRAG), they can give us some intuitions on the flexibility of our method and the optimality of its update in the subspace. We omit $\nu$ in the discussions below for the simplicity of notation.

**One-dimensional subspace** Suppose we update the variable in the one-dimensional subspace spanned by gradient direction

$$
x_t = x_{t-1} - \alpha_{1t} \cdot \hat{v}_t,
$$

where $\alpha_{1t}$ is calculated by solving the trust-region subproblem

$$\min_{\alpha_1} \left\{ \langle \hat{\boldsymbol{v}}_t, -\alpha_1 \hat{\boldsymbol{v}}_t \rangle + \frac{1}{2} \langle -\alpha_1 \hat{\boldsymbol{v}}_t, \mathrm{diag}(\sqrt{\hat{\boldsymbol{u}}_t})(-\alpha_1 \hat{\boldsymbol{v}}_t) \rangle \ \Big| \ |\alpha_1| \leq \eta \right\}.$$

In this case, the subproblem has an explicit solution $\alpha_{1t} = \min\{\eta, \frac{\langle \hat{\boldsymbol{v}}_t, \hat{\boldsymbol{v}}_t \rangle}{\langle \hat{\boldsymbol{v}}_t, \mathrm{diag}(\sqrt{\hat{\boldsymbol{u}}_t})\hat{\boldsymbol{v}}_t \rangle}\}$. As we can see from the solution, unlike SGD which adopts the learning rate set externally as the stepsize, our algorithm update the parameter with an adaptive stepsize within the learning rate. Furthermore, this adaptive stepsize is optimal according to the quadratic approximation of the loss function.

**Full-dimensional parameter space**  Suppose we update the variable along $n$ directions in the whole parameter space

$$\boldsymbol{x}_t = \boldsymbol{x}_{t-1} + \boldsymbol{p}.$$

Then trust-region subproblem becomes

$$\min_{\boldsymbol{p}} \left\{ \langle \hat{\boldsymbol{v}}_t, \boldsymbol{p} \rangle + \frac{1}{2} \langle \boldsymbol{p}, \mathrm{diag}(\sqrt{\hat{\boldsymbol{u}}_t})\boldsymbol{p} \rangle \ \Big| \ \|\boldsymbol{p}\| \leq \eta \|\hat{\boldsymbol{v}}_t\| \right\}.$$

Under this scenario, the subproblem has solution as

$$\boldsymbol{p} = -\frac{\hat{\boldsymbol{v}}_t}{\sqrt{\hat{\boldsymbol{u}}_t} + \lambda} = -\sum_{i=1}^{n} \frac{1}{\sqrt{\hat{\boldsymbol{u}}_{t,i}} + \lambda}(\hat{\boldsymbol{v}}_{t,i}\boldsymbol{e}_i),$$

where $\lambda \geq 0$ satisfies $\lambda(\|\boldsymbol{p}\| - \eta\|\hat{\boldsymbol{v}}_t\|) = 0$. When $\|\frac{\hat{\boldsymbol{v}}_t}{\sqrt{\hat{\boldsymbol{u}}_t}}\| \leq \eta\|\hat{\boldsymbol{v}}_t\|$, $\lambda = 0$ and the solution is the same as Adam's update direction (2). From the form of solution, we can see all gradient coordinates have adaptive stepsizes, which means the method optimizes the loss function along $n$ directions in the whole parameter space. Also, this update is optimal with respect to the quadratic approximation of the loss function.

## 3.2 BENEFITS OF OUR ALGORITHM

**Flexibility of update**  As in Algorithm 1, DRAG updates the variable $\boldsymbol{x}$ along EMA of gradient direction $\hat{\boldsymbol{v}}_t$ and momentum direction $\boldsymbol{d}_{t-1}$. This update direction choice acts as a trade-off between the whole space search of Adam and one direction search of SGD. Specifically, Adam adjusts stepsizes for each gradient coordinate as $\boldsymbol{x}_t = \boldsymbol{x}_{t-1} - \sum_{i=1}^{n} \frac{\eta}{\sqrt{\hat{\boldsymbol{u}}_{t,i}} + \nu}(\hat{\boldsymbol{v}}_{t,i}\boldsymbol{e}_i)$, while SGD uniformly scales each coordinate of the gradient as $\boldsymbol{x}_t = \boldsymbol{x}_{t-1} - \eta\hat{\boldsymbol{v}}_t$. DRAG, on the other hand, search along two important directions as $\boldsymbol{x}_t = \boldsymbol{x}_{t-1} - \alpha_{1t}\hat{\boldsymbol{v}}_t + \alpha_{2t}\boldsymbol{d}_t$. This choice maintains the flexibility of update direction while alleviating overfitting and excessive noises.

Moreover, the update of DRAG lies in the subspace $\mathrm{span}\{\hat{\boldsymbol{v}}_t, \boldsymbol{d}_{t-1}\} \in \mathrm{span}\{\boldsymbol{g}_0, \cdots, \boldsymbol{g}_{t-1}\}$. This means that the parameter update direction is always a combination of stochastic gradients. According to Wilson et al. (2017), this property makes DRAG always converge to the max-margin solution of the binary classification problem, which has the best generalization capacity. This helps to explain DRAG's excellent generalization performance in practice.

**Optimal stepsizes**  DRAG solves the dimension-reduced subproblem at each training epoch and finds the best update along the gradient direction and momentum direction. This optimal update is evaluated by the quadratic approximation to the loss function, where the Hessian is approximated by second moment $\sqrt{\hat{\boldsymbol{u}}_t}$ and gradient is approximated by first moment $\hat{\boldsymbol{v}}_t$. Since DRAG conducts optimal update along gradient and momentum direction within the learning rate we set, it converges faster than SGD on training dataset and is comparable with adaptive gradient methods.

**Heuristic trust-region radius**  We set the trust-region radius for the subproblem as $\eta\|\hat{\boldsymbol{v}}_t\|$. The intuition is that when gradient is large, we hope our algorithm can make a larger step to minimize the loss function significantly. While when gradient is small, we hope our method to be stable and don't change the parameters too much. This heuristic design not only frees us from changing the radius at each step as trust region method does, but also make our algorithm compatible well with dominant deep learning training pipeline without introducing extra hyperparameters. Other deep learning optimizers that adopt trust-region like framework, such as the Hessian-free optimization method (Martens et al., 2010), L-SSR1-TR (Erway et al., 2020), DRSOM (Zhang et al., 2022) introduced extra hyperparameters and may incur high extra computational overhead. From our knowledge, this is the first time that a trust-region like method is well-compatible with dominant deep learning training setting without extra hyperparameters.

# 4 CONVERGENCE ANALYSIS IN NON-CONVEX STOCHASTIC OPTIMIZATION

For the analysis of stochastic non-convex algorithm, we follow the works Zhuang et al. (2020); Guo et al. (2021) and make the following necessary definitions and also mild assumptions.

**Definition 1.** *For a differentiable function $f$, $\boldsymbol{x}$ is said to be an $\epsilon$-approximate first-order stationary point if it satisfies $\|\nabla f(\boldsymbol{x})\| \leq \epsilon$.*

**Definition 2.** *For a differentiable funtion $f(x)$, it is called L-Lipschitz smooth if it statisfies $\|\nabla f(\boldsymbol{x}) - \nabla f(\boldsymbol{y})\| \leq L\|\boldsymbol{x} - \boldsymbol{y}\|$ for a constant $L > 0$ and any $\boldsymbol{x}, \boldsymbol{y}$ in domain of $f$.*

Based on these definitions, we have the following assumption.

**Assumption 1.** *For non-convex problem $\min_{\boldsymbol{x} \in \mathbb{R}^n} f(\boldsymbol{x})$, we assume the loss $f(\boldsymbol{x})$ satisfies*

- *$f$ is L-Lipschitz smooth.*

- *The gradient estimation $\boldsymbol{g}$ is unbiased, namely $\mathbb{E}[\boldsymbol{g}_t] = \nabla f(\boldsymbol{x}_t)$, and its variance can be bounded as $\mathbb{E}[\|\boldsymbol{g}_t - \nabla f(\boldsymbol{x}_t)\|^2] \leq \sigma^2$.*

Then we can derive the convergence of our proposed algorithm and also provide its stochastic gradient complexity to find an $\epsilon$-approximate first-order stationary point.

**Theorem 1.** *Suppose Assumption 1 holds. Let $\beta_t = \beta$ and $\eta_t = \eta$ for all $t$. Assume there exist constants $\alpha, G > 0$, such that $\alpha \leq \min_t \alpha_{1t}$ and $\alpha_{1t} \leq \eta G$, $|\alpha_{2t}| \leq \eta G$. In addition, $\eta \leq \min\left\{\frac{1}{2LG}, \left(\frac{(1-\beta)^2\alpha}{8GL^2}\right)^{\frac{1}{3}}, \left(\frac{\alpha^2}{96G^2}\right)^{\frac{1}{4}}, \left(\frac{\alpha}{48LG^2}\right)^{\frac{1}{4}}, \left(\frac{\alpha}{192L^2G^3}\right)^{\frac{1}{5}}\right\}$. Then, if $1 - \beta \leq \frac{\epsilon^2}{3C_2\sigma^2}$ and $T \geq \max\left\{\frac{3C_1}{\alpha\epsilon^2}, \frac{3C_3}{(1-\beta)\epsilon^2}\right\}$, DRAG can achieve*

$$\frac{1}{T}\sum_{t=0}^{T-1}\mathbb{E}\left[\|\nabla f(\boldsymbol{x}_t)\|^2\right] \leq \epsilon^2, \quad \frac{1}{T}\sum_{t=0}^{T-1}\mathbb{E}\left[\|\boldsymbol{v}_t\|^2\right] \leq 8\epsilon^2, \tag{3}$$

*where $C_1 = 4\left(f(\boldsymbol{x}_0) - f(\boldsymbol{x}^*)\right)$, $C_2 = \frac{4\eta G}{\alpha}$ and $C_3 = \frac{2\eta G\mathbb{E}[\|\nabla f(\boldsymbol{x}_0)-(1-\beta_0)\boldsymbol{g}_0\|^2]}{\alpha}$.*

**Remark 1.** *Theorem 1 with its proof in Appendix C demonstrates that by properly selecting constant trust-region radius $\eta_t$ and constant momentum parameter $\beta_t$ (correspond to $\beta_1$ in Algorithm 1), DRAG can converge to an $\epsilon$-approximate first-order stationary point of the non-convex stochastic problem with stochastic gradient complexity $\mathcal{O}(\epsilon^{-4})$. Note that the assumptions for $\alpha_{1t}$ and $\alpha_{2t}$ are satisfied naturally with the design of DRAG, see details in Appendix B. The complexity of DRAG is of the same order as the lower bound provided by Arjevani et al. (2022). A similar complexity has also been obtained in, for example, LAMB (You et al., 2019), Adam-family (Guo et al., 2021). In the analysis of DRAG, we only need a unbiased and variance-bounded stochastic gradient, without any large mini-batch sizes requirement as in LARS (You et al., 2017) and LAMB (You et al., 2019). In addition, some previous works (Luo et al., 2018; Zaheer et al., 2018; Liu et al., 2019; Shi et al., 2020) require the momentum parameter $\beta_t$ to be very close or decreasing to zero. In contrast, DRAG requires $\beta_t$ to be close to one, which is more consistent with the practice.*

**Theorem 2.** *Suppose Assumption 1 holds. Assume there exist constants $\delta, G > 0$, such that $0 < \delta \leq \frac{\alpha_{1t}}{\eta_t} \leq G$, $\frac{|\alpha_{2t}|}{\eta_t} \leq G$. Set $\eta_t = \frac{c_\eta}{\sqrt{t+2}}$, $1 - \beta_t = \frac{Cc_\eta}{\sqrt{t+1}}$, for any $c_\eta$ and $C$ satisfying $C \geq L\sqrt{\frac{8G}{\delta}}$, and $c_\eta \leq \left\{\frac{1}{\sqrt{2}LG}, \left(\frac{\delta^2}{96G^2}\right)^{\frac{1}{2}}, \left(\frac{\delta}{48LG^2}\right)^{\frac{1}{3}}, \left(\frac{\delta}{192L^2G^3}\right)^{\frac{1}{4}}\right\}$. Then there exist two constant $C_1$ and $C_2$ which are independent with $T$, such that*

$$\frac{1}{T}\sum_{t=0}^{T-1}\mathbb{E}[\|\nabla f(\boldsymbol{x}_t)\|^2] \leq \frac{C_1}{\sqrt{T}} + \frac{C_2\log T}{\sqrt{T}}, \quad \frac{1}{T}\sum_{t=0}^{T-1}\mathbb{E}[\|\boldsymbol{v}_{t+1}\|^2] \leq \frac{8C_1}{\sqrt{T}} + \frac{8C_2\log T}{\sqrt{T}}.$$

*Given a tolerance $\epsilon > 0$, if $T \geq \tilde{\mathcal{O}}(\frac{1}{\epsilon^4})$, we have*

$$\frac{1}{T}\sum_{t=0}^{T-1}\mathbb{E}[\|\nabla f(\boldsymbol{x}_t)\|^2] \leq \epsilon^2, \quad \frac{1}{T}\sum_{t=0}^{T-1}\mathbb{E}[\|\boldsymbol{v}_{t+1}\|^2] \leq 8\epsilon^2.$$

Table 1: Top-1 test accuracy (%) of VGG16, ResNet34, DenseNet121 on CIFAR10 and CIFAR100.

|  |  | **DRAG** | SGD | Adam | AdamW | AdaBelief |
|---|---|---|---|---|---|---|
| CIFAR10 | VGG16 | **94.0** | 92.7 | 92.2 | 92.4 | 93.6 |
|  | ResNet34 | **95.6** | 94.5 | 93.3 | 94.5 | 95.4 |
|  | DenseNet121 | **96.1** | 94.5 | 93.3 | 94.6 | 95.5 |
| CIFAR100 | VGG16 | **72.8** | 69.7 | 62.2 | 68.5 | 72.2 |
|  | ResNet34 | **77.6** | 75.6 | 73.0 | 70.9 | 76.1 |
|  | DenseNet121 | **79.2** | 77.8 | 73.7 | 74.3 | 78.2 |

**Remark 2.** *Theorem 2 with its proof in Appendix D establishes an $\mathcal{O}(\log T/\sqrt{T})$ sub-linear convergence rate for DRAG by choosing a decreasing $\eta_t$ and $1 - \beta_t$ with the order $\mathcal{O}(1/\sqrt{t})$. Similar sub-linear convergence rates are also established by Zou et al. (2019) for Adam and Guo et al. (2021) for Adam-type optimizers. While Zou et al. (2019) has restrictions on the second moment momentum parameter $\beta_2$. In Theorem 2, we only need $\beta_t$ (corresponds $\beta_1$ in Algorithm 1) to increase to one.*

## 5 EXPERIMENTS

We conduct experiments on several representative benchmarks, including VGG (Simonyan & Zisserman, 2014), ResNet (He et al., 2016), DenseNet (Huang et al., 2017) on CIFAR10, CIFAR100 dataset (Krizhevsky et al., 2009), ResNet18 on ImageNet (Deng et al., 2009), and LSTM (Hochreiter & Schmidhuber, 1997) on the Penn Treebank dataset (Marcinkiewicz, 1994). We compare our algorithm DRAG with some popular deep learning optimizers, including SGD (Robbins & Monro, 1951), Adam (Kingma & Ba, 2014), AdamW (Loshchilov & Hutter, 2018), AdaBound (Luo et al., 2018), AdaBelief (Zhuang et al., 2020), RAdam (Liu et al., 2019), Yogi (Zaheer et al., 2018), and Padam (Chen et al., 2021). Experimental results show that DRAG has faster convergence speed compared with SGD and it achieves state-of-the-art generalization performance. We also conduct ablation study to show 1) two search directions (DRAG) performs better than one direction and multiple directions and 2) DRAG is robust to different learning rate schedules. At the end of ablation study, we give some advice for practitioners to use DRAG.

### 5.1 CNNs ON IMAGE CLASSIFICATION

We conducted experiments for VGG16 with Batch Normalization, ResNet34, and DenseNet121 on CIFAR10 an CIFAR100 dataset. The experimental setting is borrowed from AdaBelief (Zhuang et al., 2020) and we also use their default setting for all the hyperparameters. For DRAG, we choose its learning rate to be the same as in SGD, which is 0.1, and weight decay factor is 0.0015 for CIFAR10 and 0.0025 for CIFAR100. Other hyperparameters of DRAG is the same as the default setting ($\beta_1 = 0.9$, $\beta_2 = 0.999$, $\epsilon = 10^{-8}$). As Figure 1 shows, DRAG has convergence speed comparable with adaptive gradient methods and it attains the best generalization performance. To be specific, DRAG obtains more than 0.5% generalization accuracy gain over AdaBelief (Zhuang et al., 2020) on most tasks. The detailed test accuracy is summarized in Table 1.

We also train ResNet18 on ImageNet under the official training setting in He et al. (2016), and compare the top-1 test accuracy of DRAG with the best result of other optimizers in the literature. Experimental results show DRAG has the best generalization capacity. Details are in Table 2.

The possible reasons for this improvement on the convergence speed and generalization capacity is 1) DRAG searches for the optimal update along two directions and thus converges faster, 2) DRAG confines the search of update within the two-dimensional subspace spanned by gradient and momentum direction to avoid overfitting and alleviating the influence of noises, therefore it generalizes better.

### 5.2 LSTMs ON LANGUAGE MODELING

We experimented with LSTM on the Penn Treebank dataset and record the perplexity (lower is better). We follow the exact experimental setting in Adabelief (Zhuang et al., 2020) and use their

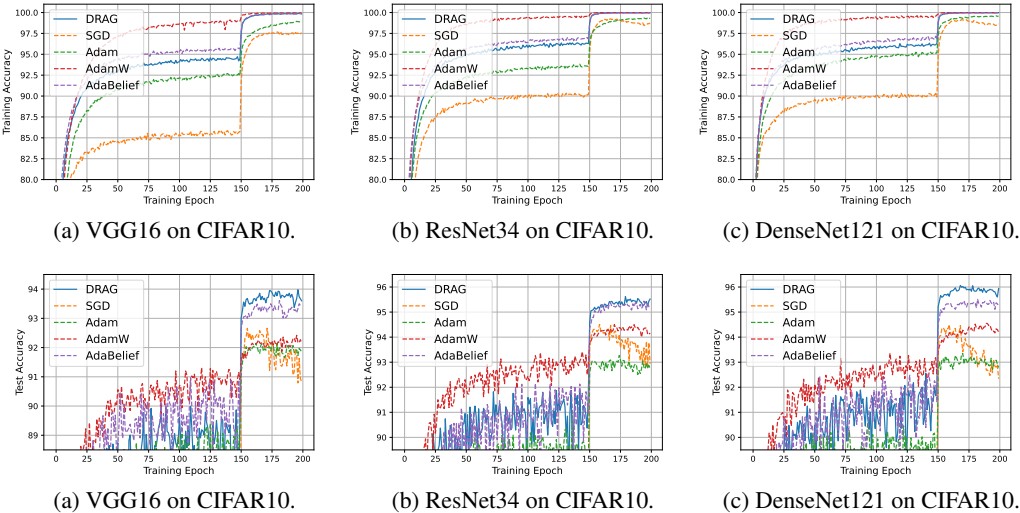

(a) VGG16 on CIFAR10.     (b) ResNet34 on CIFAR10.     (c) DenseNet121 on CIFAR10.

(a) VGG16 on CIFAR10.     (b) ResNet34 on CIFAR10.     (c) DenseNet121 on CIFAR10.

Figure 1: Training and test accuracy of CNNs on CIFAR10 dataset.

Table 2: Top-1 test accuracy (%) of ResNet18 on ImageNet. All results except DRAG are reported by Zhuang et al. (2020) and Chen et al. (2021).

| DRAG | SGD | Adam | AdamW | AdaBound | Padam | AdaBelief |
|------|------|------|-------|----------|-------|-----------|
| 70.41 | 70.23 | 63.79 | 67.93 | 68.13 | 70.07 | 70.08 |

default hyperparameters except for SGD. For SGD, we use the same hyperparameters as DRAG to make a fair comparison between the two. For SGD and DRAG, we set their learning rate as 25, 75, 75 for 1,2,3-layer LSTM and weight decay factor as $2.5 \times 10^{-6}$. SGD's generalization performance in our setting is better than the results provided by Zhuang et al. (2020). As show in Figure 2, for 1-layer, 2-layer, and 3-layer LSTM, DRAG's convergence speed is faster than SGD and comparable to adaptive gradient methods. From Table 3, we can see that DRAG attains more than 0.5 less perplexity than other optimizers. The fast convergence speed may be attributed to the optimal update DRAG takes and the good generalization performance may be due to DRAG's two-direction search. The gradient direction inherits SGD's good generalization property and the extra momentum direction further improves its performance.

### 5.3 ABLATION STUDY

**Different search directions** We compare the performance of algorithms that solve the trust-region subproblem in one-dimensional, two-dimensional (DRAG), and three-dimensional subspaces as described in Section 3.1. As show in Table 4 in Appendix E, DRAG generalizes better than its one search direction and three search direction counterparts. The reason is that DRAG updates in more directions than the one search direction counterpart while its subproblem can be solved more accurately than the three direction counterpart, since low-dimensional subproblem can be solved with less numerical errors in single precision arithmetic by GPU.

**Robustness to learning rate schedule** DRAG is robust to different choices of learning rate schedule. Except for letting the learning rate decay at epoch 150 as in Section 5.1, we also conduct experiments on decaying the learning rate at epoch 120 and adopting cosine annealing learning rate schedule. The only change of hyperparameter setting from Section 5.1 is we increase the learning rate of DRAG from 0.1 to 0.12 in cosine annealing schedule. The intuition is that when the trust-region radius is decreased during the training process, we need a larger initial radius to converge to a better local minima. We compared DRAG's test performance with other optimizers with VGG16 on CIFAR10, details are presented in Table 5 in Appendix E, which shows that DRAG enjoys the best generalization performance for all the learning rate schedules.

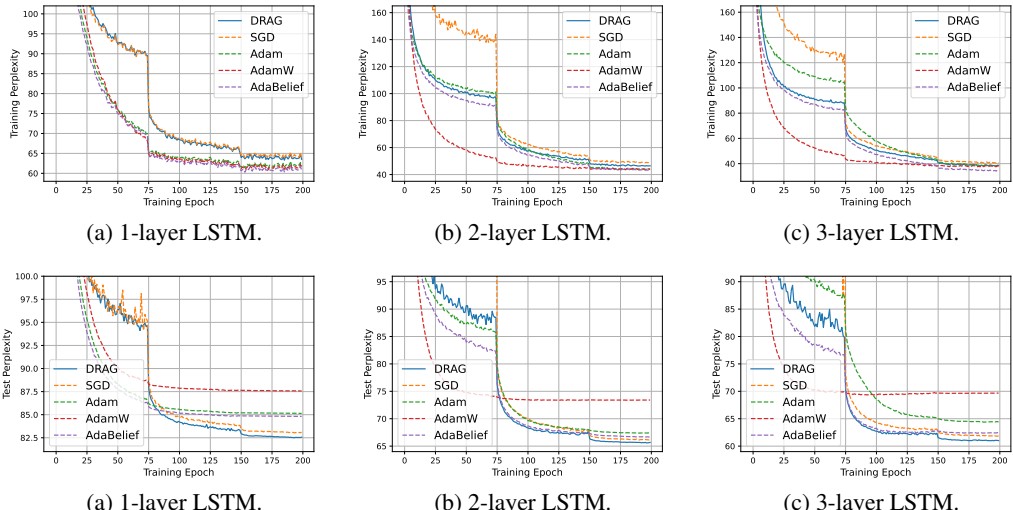

Figure 2: Training and test perplexity of 1,2,3-layer LSTM on Penn Treebank dataset.

Table 3: Test perplexity (lower is better) of 1-layer, 2-layer, and 3-layer LSTM on PTB dataset. All results except DRAG, SGD, and Padam are reported by Adabelief (Zhuang et al., 2020).

|  | **DRAG** | SGD | AdaBound | Adam | AdamW | AdaBelief | RAdam | Yogi | Padam |
|---|---|---|---|---|---|---|---|---|---|
| 1-layer | **82.5** | 83.0 | 84.3 | 85.1 | 87.7 | 84.8 | 86.5 | 86.5 | 84.0 |
| 2-layer | **65.6** | 66.1 | 67.5 | 67.4 | 72.8 | 66.3 | 72.3 | 71.3 | 66.3 |
| 3-layer | **61.0** | 61.8 | 63.6 | 64.3 | 69.9 | 61.8 | 70.0 | 67.5 | 62.8 |

For practitioners, any task that can use SGD can use DRAG to achieve faster convergence and comparable generalization performance with negligible extra computational overhead. The user only needs to set the learning rate the same as in SGD or slightly larger. For a new task, if one values good generalization performance, one can always use DRAG instead of SGD to enjoy easier hyperparameter tunning. DRAG is more robust than SGD when large learning rate is used. For instance, when training VGG16 on CIFAR10 dataset, setting the learning rate to 0.5 still allows DRAG to attain over 90 percent test accuracy, but SGD diverge and fail in the training process.

## 6 CONCLUSION

In this paper we propose the DRAG algorithm, which finds the optimal update of the parameters along gradient and momentum directions at each iteration. Compared with Adam, DRAG reduces the flexibility of update direction from searching in the whole parameter space to updating in a two-dimensional subspace: therefore is less susceptible to overfitting and has better generalization performance. Compared with SGD, DRAG inherits the gradient update direction and also update along an extra momentum direction, thus it has faster convergence speed and comparable generalization capacity. Theoretically we prove that DRAG has the same order of stochastic gradient complexity as the lower bound for non-convex stochastic optimization (Arjevani et al., 2022). Experimentally we show that DRAG has faster convergence speed compared with SGD and it attains state-of-the-art generalization performance. Our algorithm can be further generalized to any number of search directions and any choice of Hessian approximation.

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

## A    SOLVE THE TRUST-REGION SUBPROBLEM

Recall the trust region subproblem

$$\min_{\alpha} \quad \langle \alpha, C_t \rangle + \frac{1}{2}\langle \alpha, Q_t \alpha \rangle$$

$$\text{s.t.} \quad \sqrt{\langle \alpha, G_t \alpha \rangle} \le \eta \|\hat{v}_t\|,$$

where $\alpha := \begin{bmatrix} \alpha_1 \\ \alpha_2 \end{bmatrix}$, $C_t := \begin{bmatrix} -\hat{v}_t^T \hat{v}_t \\ \hat{v}_t^T d_{t-1} \end{bmatrix}$, $Q_t := \begin{bmatrix} \hat{v}_t^T H_t \hat{v}_t & -\hat{v}_t^T H_t \\ -d_{t-1}^T H_t \hat{v}_t & d_{t-1}^T H_t d_{t-1} \end{bmatrix}$, and $G_t := \begin{bmatrix} \hat{v}_t^T \hat{v}_t & -\hat{v}_t^T d_{t-1} \\ -d_{t-1}^T \hat{v}_t & d_{t-1}^T d_{t-1} \end{bmatrix}$, $H_t = \text{diag}(\sqrt{\hat{u}_t} + \nu)$.

In order to solve this trust region subproblem, we transform it into a standard trust region subproblem with $L_2$-norm constraint.

When matrix $G_t$ is positive definite, we have

$$G_t = L_t L_t^T \text{ (Cholesky Decomposition)}$$

$$\sqrt{\alpha^T G_t \alpha} = \sqrt{(L_t^T \alpha)^T L_t^T \alpha} = \|L_t^T \alpha\| \le \eta \|\hat{v}_t\|.$$

So we let $y = L_t^T \alpha$, then $\alpha = L_t^{-T} y$ and the subproblem becomes

$$\min_{y} \quad \langle C_t, L_t^{-T} y \rangle + \frac{1}{2}\langle L_t^{-T} y, Q_t L_t^{-T} y \rangle$$

$$\text{s.t.} \quad \|y\| \le \eta \|\hat{v}_t\|$$

$$\iff \quad \min_{y} \quad \langle L_t^{-1} C_t, y \rangle + \frac{1}{2}\langle y, L_t^{-1} Q_t L_t^{-T} y \rangle$$

$$\text{s.t.} \quad \|y\| \le \eta \|\hat{v}_t\|.$$

In this way, the trust region subproblem is transformed to a standard spherical constrained quadratic optimization problem and it can be solved efficiently (Wright et al., 1999).

When $|G_t| = 0$, this means $\hat{v}_t$ is linearly dependent with $d_{t-1}$. In this case, we solve the one-dimensional subproblem as described in Section 3.1.

## B    HOW $\alpha_{1t}, \alpha_{2t}$ SATISFY THE ASSUMPTIONS NATURALLY

The trust-region subproblem to be solved in Algorithm 1 has global optimality condition (Luenberger et al., 1984) given by

$$\begin{cases} (Q_t + \lambda G_t)\alpha + C_t = 0 \\ Q_t + \lambda G_t \succeq 0 \\ \lambda(\|\alpha\|_{G_t} - \eta \|\hat{v}_t\|) = 0, \quad \lambda \ge 0. \end{cases}$$

By its construction, we know that $G_t$ is positive semidefinite. In practice, numerical issues sometimes make it indefinite, leaving the trust-region subproblem insoluble. Thus, we make an adjustment to $G_t$

$$G_t = \begin{cases} G_t & \text{if } \lambda_{min} \ge \varepsilon_0 \text{ or } |G_t| = 0 \\ \varepsilon_0 I & \text{o.w.} \end{cases}$$

where $\lambda_{min}$ is the smallest eigenvalue of $G_t$. In this way, when $|G_t| \ne 0$, we have

$$\|\alpha\| \le \|G_t^{-1/2}\| \|\alpha\|_{G_t} \le \eta \|G_t^{-1/2}\| \|\hat{v}_t\| \le \eta \frac{\|\hat{v}_t\|}{\sqrt{\varepsilon_0}},$$

which means

$$|\frac{\alpha_{1t}}{\eta}|, |\frac{\alpha_{2t}}{\eta}| \le \frac{\|\hat{v}_t\|}{\sqrt{\varepsilon_0}}.$$

With the common additional assumption that stochastic gradient $g_t = \tilde{\nabla} f(x_t)$ has bounded $L_\infty$ norm, i.e. $\|g_t\|_\infty \leq G_\infty$, then $\hat{v}_t$ as an moving average of $g_t$ also has bounded norm $\|\hat{v}_t\|$. Therefore, we can see that $|\frac{\alpha_{1t}}{\eta}|, |\frac{\alpha_{2t}}{\eta}|$ are upper bounded by a constant.

When $|G_t| = 0$, which means $d_{t-1}$ is parallel with $\hat{v}_t$. Then we only need to find the optimal update within the trust-region along gradient direction $\hat{v}_t$. In this case, we manually set $\alpha_{2t} = 0$ in our implementation of DRAG, and then $\alpha_1$ satisfies $|\alpha_1| \leq \eta$.

From discussions above, we can see the assumption that $|\frac{\alpha_{1t}}{\eta}|, |\frac{\alpha_{2t}}{\eta}|$ are upper bounded in Theorem 1 and Theorem 2 is satisfied given the common assumption that stochastic gradient $g_t = \tilde{\nabla} f(x_t)$ has bounded $L_\infty$ norm. For the simplicity of notations, we directly make assumptions for $\alpha_{1t}$ and $\alpha_{2t}$ in Theorem 1 and Theorem 2.

For the assumption that $\alpha_{1t}$ is positive and $\frac{\alpha_{1t}}{\eta}$ is lower bounded by a constant, we give an explanation here by intuition and empirical results. Gradient direction is what we considered the most important update direction locally, because by the training pipeline of neural networks, stochastic gradients of training parameters are the new information we gain at each iteration. Thus, we consider the update should at least move towards the gradient descent direction rather than move towards the gradient ascent direction. Moreover, from the observations of $\alpha_{1t}$ under all the experimental settings, $\alpha_{1t}$ is always positive and $\frac{\alpha_{1t}}{\eta}$ is always larger than 0.1. Therefore, this assumption on $\alpha_{1t}$ is reasonable based on common sense and holds true in practice.

## C  PROOF OF THEOREM 1

One key ingredient in our analysis is an existing variance recursion of the stochastic estimator based on moving average, which is given by the following lemma.

**Lemma 3** (Variance Recursion (Wang et al., 2017)). *Suppose Assumption 1 holds, then we have*

$$\mathbb{E}_t[\|v_{t+1} - \nabla f(x_t)\|^2] \leq \beta \|v_t - \nabla f(x_{t-1})\|^2 + 2(1-\beta)^2 \mathbb{E}_t[\|g_t - \nabla f(x_t)\|^2] + \frac{L^2\|d_t\|^2}{1-\beta},$$

*where $\mathbb{E}_t[\cdot]$ denotes the conditional expectation with respect to all randomness before $g_t$.*

Before proving Theorem 1, we need to prove the following auxiliary lemma.

**Lemma 4.** *Suppose Assumption 1 holds. Assume there exist $\alpha, \eta, \delta, G > 0$, such that $\alpha \leq \min_t \alpha_{1t}$, $\max_t \eta_t \leq \eta$, and $0 < \delta \leq \frac{\alpha}{\eta} \leq \frac{\alpha_{1t}}{\eta_t} \leq G$, $\frac{|\alpha_{2t}|}{\eta_t} \leq G$, $(\delta, G)$ are constants independent with $t$. In addition, $\eta \leq \min\left\{\frac{1}{2LG}, \frac{1-\beta}{2L}\sqrt{\frac{\delta}{2G}}, \frac{\delta}{4\sqrt{6}G}, \left(\frac{\delta}{48LG^2}\right)^{\frac{1}{3}}, \left(\frac{\delta}{192L^2G^3}\right)^{\frac{1}{4}}\right\}$. Then there exist positive constants $C_1$, $C_2$ and $C_3$, which are all independent with $T$, such that the following estimation holds:*

$$\frac{1}{T}\sum_{t=0}^{T-1}\mathbb{E}\left[\|\nabla f(x_t)\|^2\right] \leq \frac{C_1}{T\alpha} + C_2(1-\beta)\sigma^2 + \frac{C_3}{T(1-\beta)},$$

$$\frac{1}{T}\sum_{t=0}^{T-1}\mathbb{E}\left[\|v_t\|^2\right] \leq \frac{8C_1}{T\alpha} + 8C_2(1-\beta)\sigma^2 + \frac{8C_3}{T(1-\beta)}.$$

(4)

*Proof.* Since $F$ is $L$-smooth, we have

$$f(x_{t+1}) \leq f(x_t) + \langle \nabla f(x_t), -\alpha_{1t}v_{t+1} + \alpha_{2t}d_t \rangle + \frac{L}{2}\| -\alpha_{1t}v_{t+1} + \alpha_{2t}d_t\|^2$$

$$= f(x_t) - \alpha_{1t}\langle \nabla f(x_t), v_{t+1} \rangle + \alpha_{2t}\langle \nabla f(x_t), d_t \rangle + \frac{L\alpha_{1t}^2}{2}\|v_{t+1}\|^2 + \frac{L\alpha_{2t}^2}{2}\|d_t\|^2 - L\alpha_{1t}\alpha_{2t}\langle v_{t+1}, d_t \rangle$$

$$= f(x_t) + \frac{\alpha_{1t}}{2}\|\nabla f(x_t) - v_{t+1}\|^2 - \frac{\alpha_{1t}(1-L\alpha_{1t})}{2}\|v_{t+1}\|^2 - \frac{\alpha_{1t}}{2}\|\nabla f(x_t)\|^2 + \alpha_{2t}\langle \nabla f(x_t), d_t \rangle$$

$$+ \frac{L\alpha_{2t}^2}{2}\|d_t\|^2 - L\alpha_{1t}\alpha_{2t}\langle v_{t+1}, d_t \rangle.$$

(5)

By Lemma 3, we can obtain

$$\sum_{t=1}^{T}\mathbb{E}[\|\nabla f(x_{t-1})-v_t\|^2]\leq\frac{1}{1-\beta}\mathbb{E}[\|\nabla f(x_0)-v_1\|^2]+2(1-\beta)T\sigma^2+\frac{L^2}{(1-\beta)^2}\mathbb{E}\left[\sum_{t=1}^{T}\|d_t\|^2\right].$$
(6)

Taking expectation for both sides of (5) and taking summation among $t=0,...,T-1$, combining with (6), we have

$$\mathbb{E}\left[f(x_T)-f(x_0)\right]$$

$$\leq\frac{\eta G}{2}\left[\frac{\mathbb{E}[\|\nabla f(x_0)-v_1\|^2]}{1-\beta}+2(1-\beta)T\sigma^2+\frac{L^2}{(1-\beta)^2}\sum_{t=1}^{T}\mathbb{E}[\|d_t\|^2]\right]-\sum_{t=0}^{T-1}\frac{\alpha_{1t}}{2}\mathbb{E}[\|\nabla f(x_t)\|^2]$$

$$-\sum_{t=0}^{T-1}\frac{\alpha_{1t}(1-L\alpha_{1t})}{2}\mathbb{E}[\|v_{t+1}\|^2]+\sum_{t=0}^{T-1}\left(\mathbb{E}[\alpha_{2t}\langle\nabla f(x_t),d_t\rangle]+\frac{L\alpha_{2t}^2}{2}\|d_t\|^2-\mathbb{E}[L\alpha_{1t}\alpha_{2t}\langle v_{t+1},d_t\rangle]\right).$$

By AM-GM inequality,

$$\alpha_{2t}\langle\nabla f(x_t),d_t\rangle\leq\frac{\alpha_{1t}}{4}\|\nabla f(x_t)\|^2+\frac{\alpha_{2t}^2}{\alpha_{1t}}\|d_t\|^2,$$

$$-L\alpha_{1t}\alpha_{2t}\langle v_{t+1},d_t\rangle\leq\frac{\alpha_{1t}(1-L\alpha_{1t})}{4}\|v_{t+1}\|^2+\frac{L^2\alpha_{1t}\alpha_{2t}^2}{1-L\alpha_{1t}}\|d_t\|^2.$$
(7)

Combining all together, we have

$$\sum_{t=0}^{T-1}\frac{\alpha_{1t}}{4}\mathbb{E}[\|\nabla f(x_t)\|^2]$$

$$\leq f(x_0)-f(x^*)+\frac{\eta G}{2(1-\beta)}\mathbb{E}[\|\nabla f(x_0)-v_1\|^2]+\sum_{t=1}^{T}\frac{\eta GL^2}{2(1-\beta)^2}\mathbb{E}[\|d_t\|^2]$$

$$+\eta G(1-\beta)T\sigma^2+\sum_{t=0}^{T-1}\left(\frac{\alpha_{2t}^2}{\alpha_{1t}}+\frac{L\alpha_{2t}^2}{2}+\frac{L^2\alpha_{1t}\alpha_{2t}^2}{1-L\alpha_{1t}}\right)\mathbb{E}[\|d_t\|^2]-\sum_{t=0}^{T-1}\frac{\alpha_{1t}(1-L\alpha_{1t})}{4}\mathbb{E}[\|v_{t+1}\|^2],$$
(8)

where $x^*$ is one of the global minimizer of $F$. Since $\alpha_{1t}\leq\eta G\leq\frac{1}{2L}$, we have $\frac{\alpha_{1t}(1-L\alpha_{1t})}{4}\geq\frac{\alpha_{1t}}{8}$. By the conditions for $\eta$ and $\alpha$, we have $\frac{\alpha_{1t}}{16}\geq\frac{\alpha}{16}\geq\frac{\eta^3 GL^2}{2(1-\beta)^2}\geq\frac{\eta GL^2\eta_{t+1}^2}{2(1-\beta)^2}$, $\frac{\alpha_{1t}}{96}\geq\frac{\alpha}{96}\geq\frac{\eta^4 G^2}{\alpha}\geq\frac{\alpha_{2,t+1}^2\eta_{t+1}^2}{\alpha_{1,t+1}}$, $\frac{\alpha_{1t}}{96}\geq\frac{\alpha}{96}\geq\frac{L\eta^4 G^2}{2}\geq\frac{L\alpha_{2,t+1}^2\eta_{t+1}^2}{2}$, and $\frac{\alpha_{1t}}{96}\geq\frac{\alpha}{96}\geq 2L^2\eta^5 G^3\geq\frac{L^2\alpha_{1,t+1}\alpha_{2,t+1}^2\eta_{t+1}^2}{1-L\alpha_{1,t+1}}$. By $\|d_t\|\leq\eta_t\|v_t\|$. Since $v_0=0$, we have

$$\sum_{t=0}^{T-1}\left(\frac{\alpha_{2t}^2}{\alpha_{1t}}+\frac{L\alpha_{2t}^2}{2}+\frac{L^2\alpha_{1t}\alpha_{2t}^2}{1-L\alpha_{1t}}\right)\|d_t\|^2+\sum_{t=1}^{T}\frac{\eta GL^2}{2(1-\beta)^2}\|d_t\|^2-\sum_{t=0}^{T-1}\frac{\alpha_{1t}(1-L\alpha_{1t})}{4}\|v_{t+1}\|^2$$

$$\leq-\frac{\alpha}{8}\sum_{t=0}^{T-1}\|v_{t+1}\|^2+\sum_{t=0}^{T-1}\left(\frac{\alpha_{2t}^2\eta_t^2}{\alpha_{1t}}+\frac{L\alpha_{2t}^2\eta_t^2}{2}+\frac{L^2\alpha_{1t}\alpha_{2t}^2\eta_t^2}{1-L\alpha_{1t}}\right)\|v_t\|^2+\sum_{t=0}^{T-1}\frac{\eta GL^2\eta_{t+1}^2}{2(1-\beta)^2}\|v_{t+1}\|^2$$

$$\leq-\frac{\alpha}{32}\sum_{t=0}^{T-1}\|v_{t+1}\|^2.$$
(9)

Combining (8) and (9), we can obtain

$$\sum_{t=0}^{T-1}\frac{\alpha_{1t}}{4}\mathbb{E}[\|\nabla f(x_t)\|^2]\leq f(x_0)-f(x^*)+\frac{\eta G}{2(1-\beta)}\mathbb{E}[\|\nabla f(x_0)-v_1\|^2]+\eta G(1-\beta)T\sigma^2,$$

$$\frac{\alpha}{32}\sum_{t=0}^{T-1}\mathbb{E}[\|v_{t+1}\|^2]\leq f(x_0)-f(x^*)+\frac{\eta G}{2(1-\beta)}\mathbb{E}[\|\nabla f(x_0)-v_1\|^2]+\eta G(1-\beta)T\sigma^2.$$

Dividing the above two inequalities by $\frac{\alpha T}{4}$ and $\frac{\alpha T}{32}$ respectively, we have

$$\frac{1}{T}\sum_{t=0}^{T-1}\mathbb{E}[\|\nabla f(x_t)\|^2] \leq \frac{4\left(f(x_0)-f(x^*)\right)}{T\alpha}+\frac{2G\mathbb{E}[\|\nabla f(x_0)-v_1\|^2]}{\delta(1-\beta)T}+\frac{4G(1-\beta)\sigma^2}{\delta},$$

$$\frac{1}{T}\sum_{t=0}^{T-1}\mathbb{E}[\|v_{t+1}\|^2] \leq \frac{32\left(f(x_0)-f(x^*)\right)}{T\alpha}+\frac{16G\mathbb{E}[\|\nabla f(x_0)-v_1\|^2]}{\delta(1-\beta)T}+\frac{32G(1-\beta)\sigma^2}{\delta},$$

which completes the proof by letting $C_1 = 4\left(f(x_0)-f(x^*)\right)$, $C_2 = \frac{4G}{\delta}$, $C_3 = \frac{2G\mathbb{E}[\|\nabla f(x_0)-v_1\|^2]}{\delta}$. $\square$

**Proof of Theorem 1**

*Proof.* By the selections of $\alpha$ and $\eta_t$ in Theorem 1, let $\delta = \alpha/\eta$. By Lemma 4, we have

$$\frac{1}{T}\sum_{t=0}^{T-1}\mathbb{E}\left[\|\nabla f(x_t)\|^2\right] \leq \frac{C_1}{T\alpha}+C_2(1-\beta)\sigma^2+\frac{C_3}{T(1-\beta)}.$$

The conditions $1-\beta \leq \frac{\epsilon^2}{3C_2\sigma^2}$ and $T \geq \max\left\{\frac{3C_1}{\alpha\epsilon^2}, \frac{3C_3}{(1-\beta)\epsilon^2}\right\}$ lead to $\frac{C_1}{T\alpha} \leq \frac{\epsilon^2}{3}$, $C_2(1-\beta)\sigma^2 \leq \frac{\epsilon^2}{3}$, $\frac{C_3}{T(1-\beta)} \leq \frac{\epsilon^2}{3}$. This completes the proof. $\square$

## D    PROOF OF THEOREM 2

*Proof.* From (5) in Lemma 4, we have

$$f(x_{t+1}) \leq f(x_t) + \frac{\alpha_{1t}}{2}\|\nabla f(x_t)-v_{t+1}\|^2 - \frac{\alpha_{1t}(1-L\alpha_{1t})}{2}\|v_{t+1}\|^2 - \frac{\alpha_{1t}}{2}\|\nabla f(x_t)\|^2 + \alpha_{2t}\langle\nabla f(x_t), d_t\rangle$$
$$+ \frac{L\alpha_{2t}^2}{2}\|d_t\|^2 - L\alpha_{1t}\alpha_{2t}\langle v_{t+1}, d_t\rangle. \tag{10}$$

By Lemma 3, we have

$$(1-\beta_t)\|v_t-\nabla f(x_{t-1})\|^2 \leq \|v_t-\nabla f(x_{t-1})\|^2 - \mathbb{E}_t[\|v_{t+1}-\nabla f(x_t)\|^2] + 2(1-\beta_t)^2\mathbb{E}_t[\|\nabla f(x_t)-g_t\|^2] + \frac{L^2\|d_t\|^2}{1-\beta_t}.$$

Taking expectation and summation for $t = 1, ..., T$, we get

$$\sum_{t=0}^{T-1}\mathbb{E}[(1-\beta_{t+1})\|\nabla f(x_t)-v_{t+1}\|^2] \leq \mathbb{E}[\|v_1-\nabla f(x_0)\|^2] + \sum_{t=1}^{T}\left(2(1-\beta_t)^2\sigma^2 + \frac{L^2\mathbb{E}[\|d_t\|^2]}{1-\beta_t}\right). \tag{11}$$

Note that $1-\beta_{t+1} = C\eta_t$, so $\frac{\alpha_{1t}}{2} \leq \frac{G}{2}\eta_t = \frac{G}{2C}(1-\beta_{t+1})$. Taking expectation for both sides of (10) and taking summation among $t = 0, ..., T-1$, combining with (11), we can obtain

$$\mathbb{E}\left[f(x_T)-f(x_0)\right]$$
$$\leq \frac{G}{2C}\left[\mathbb{E}[\|v_1-\nabla f(x_0)\|^2] + \sum_{t=1}^{T}\left(2(1-\beta_t)\sigma^2 + \frac{L^2\|d_t\|^2}{1-\beta_t}\right)\right] - \sum_{t=0}^{T-1}\frac{\alpha_{1t}(1-L\alpha_{1t})}{2}\mathbb{E}[\|v_{t+1}\|^2]$$
$$- \sum_{t=0}^{T-1}\frac{\alpha_{1t}}{2}\mathbb{E}[\|\nabla f(x_t)\|^2] + \sum_{t=0}^{T-1}\left(\mathbb{E}[\alpha_{2t}\langle\nabla f(x_t), d_t\rangle] + \frac{L\alpha_{2t}^2}{2}\|d_t\|^2 - \mathbb{E}[L\alpha_{1t}\alpha_{2t}\langle v_{t+1}, d_t\rangle]\right). \tag{12}$$

From (7) and (12), we can get

$$
\sum_{t=0}^{T-1} \frac{\alpha_{1t}}{4} \mathbb{E}[\|\nabla f(x_t)\|^2]
$$

$$
\leq f(x_0) - f(x^*) + \frac{G}{2C} \mathbb{E}[\|\nabla f(x_0) - v_1\|^2] + \sum_{t=1}^{T} \frac{G}{C}(1 - \beta_t)^2 \sigma^2 + \sum_{t=1}^{T} \frac{GL^2 \|d_t\|^2}{2C(1 - \beta_t)} \quad (13)
$$

$$
+ \sum_{t=0}^{T-1} \left( \frac{\alpha_{2t}^2}{\alpha_{1t}} + \frac{L\alpha_{2t}^2}{2} + \frac{L^2 \alpha_{1t} \alpha_{2t}^2}{1 - L\alpha_{1t}} \right) \mathbb{E}[\|d_t\|^2] - \sum_{t=0}^{T-1} \frac{\alpha_{1t}(1 - L\alpha_{1t})}{4} \mathbb{E}[\|v_{t+1}\|^2],
$$

By the conditions for $c_\eta$ and $C$, we have $\alpha_{1t} \leq \eta_t G \leq \frac{1}{2L}$, $\frac{\alpha_{1t}(1-L\alpha_{1t})}{4} \geq \frac{\alpha_{1t}}{8}$. By similar arguments in the proof of Lemma 4, we have $\frac{\alpha_{1t}}{16} \geq \frac{\delta\eta_t}{16} \geq \frac{\eta_{t+1}^2 GL^2}{2\eta_t C^2} = \frac{GL^2 \eta_{t+1}^2}{2(1-\beta_{t+1})C}$, $\frac{\alpha_{1t}}{96} \geq \frac{\delta\eta_t}{96} \geq \frac{\alpha_{2,t+1}^2 \eta_{t+1}^2}{\alpha_{1,t+1}}$, $\frac{\alpha_{1t}}{96} \geq \frac{\delta\eta_t}{96} \geq \frac{L\alpha_{2,t+1}^2 \eta_{t+1}^2}{2}$, and $\frac{\alpha_{1t}}{96} \geq \frac{\delta\eta_t}{96} \geq 2L^2 \eta_{t+1}^5 G^3 \geq \frac{L^2 \alpha_{1,t+1} \alpha_{2,t+1}^2 \eta_{t+1}^2}{1-L\alpha_{1,t+1}}$. By $\|d_t\| \leq \eta_t \|v_t\|$. Since $v_0 = 0$, we can get

$$
\sum_{t=0}^{T-1} \left( \frac{\alpha_{2t}^2}{\alpha_{1t}} + \frac{L\alpha_{2t}^2}{2} + \frac{L^2 \alpha_{1t} \alpha_{2t}^2}{1 - L\alpha_{1t}} \right) \|d_t\|^2 + \sum_{t=1}^{T} \frac{GL^2}{2C(1 - \beta_t)} \|d_t\|^2 - \sum_{t=0}^{T-1} \frac{\alpha_{1t}(1 - L\alpha_{1t})}{4} \|v_{t+1}\|^2
$$

$$
\leq - \sum_{t=0}^{T-1} \frac{\alpha_{1t}}{8} \|v_{t+1}\|^2 + \sum_{t=0}^{T-1} \left( \frac{\alpha_{2t}^2 \eta_t^2}{\alpha_{1t}} + \frac{L\alpha_{2t}^2 \eta_t^2}{2} + \frac{L^2 \alpha_{1t} \alpha_{2t}^2 \eta_t^2}{1 - L\alpha_{1t}} \right) \|v_t\|^2 + \sum_{t=0}^{T-1} \frac{GL^2 \eta_{t+1}^2}{2C(1 - \beta_{t+1})} \|v_{t+1}\|^2
$$

$$
\leq - \sum_{t=0}^{T-1} \frac{\alpha_{1t}}{32} \|v_{t+1}\|^2.
$$

$$(14)$$

Combining (13) and (14), we can obtain

$$
\sum_{t=0}^{T-1} \frac{\delta\eta_t}{4} \mathbb{E}[\|\nabla f(x_t)\|^2] \leq \sum_{t=0}^{T-1} \frac{\alpha_{1t}}{4} \mathbb{E}[\|\nabla f(x_t)\|^2] \leq f(x_0) - f(x^*) + \frac{G\mathbb{E}[\|\nabla f(x_0) - v_1\|^2]}{2C} + \sum_{t=1}^{T} \frac{G\sigma^2}{C}(1 - \beta_t)^2,
$$

$$
\sum_{t=0}^{T-1} \frac{\delta\eta_t}{32} \mathbb{E}[\|v_{t+1}\|^2] \leq \sum_{t=0}^{T-1} \frac{\alpha_{1t}}{32} \mathbb{E}[\|v_{t+1}\|^2] \leq f(x_0) - f(x^*) + \frac{G\mathbb{E}[\|\nabla f(x_0) - v_1\|^2]}{2C} + \sum_{t=1}^{T} \frac{G\sigma^2}{C}(1 - \beta_t)^2.
$$

Then, the final assertion can be obtained by $\sum_{t=1}^{T} \frac{1}{t+1} = \mathcal{O}(\log T)$. This completes the proof. $\square$

## E  EXPERIMENTAL RESULTS FOR ABLATION STUDY

Table 4: Test accuracy of algorithms solving the trust-region subproblem with one, two, and three search directions on CIFAR10.

|  | VGG16 | ResNet34 | DenseNet121 |
|---|---|---|---|
| 1 direction | 93.8 | 95.3 | 96.0 |
| DRAG | **94.0** | **95.6** | **96.1** |
| 3 directions | 93.8 | 95.4 | 95.7 |

Table 5: Test accuracy of VGG16 on CIFAR-10 with three different learning rate schedules.

|  | **DRAG** | SGD | Adam | AdamW | Adabelief |
|---|---|---|---|---|---|
| Cosine Annealing | **94.3** | 94.0 | 92.2 | 92.4 | 94.1 |
| Decay at 120 epoch | **93.8** | 92.5 | 91.8 | 92.6 | 93.6 |
| Decay at 150 epoch | **94.0** | 92.7 | 92.2 | 92.4 | 93.6 |

