# OpenReview forum: "DIMENSION-REDUCED ADAPTIVE GRADIENT METHOD"
_ICLR.cc/2023/Conference — Submitted to ICLR 2023_

### Official Review · Reviewer_Ymav · 2022-10-23

**Confidence:** 4
**Correctness:** 3
**Technical Novelty And Significance:** 2
**Empirical Novelty And Significance:** 3
**Recommendation:** 5

**Clarity, Quality, Novelty And Reproducibility:**

Though this paper needs clarifications, the overall clarity is good. Since it is inspired by the prior method DRSOM (Zhang et al., 2022), the novelty is not really impressive.

**Strength And Weaknesses:**

The strengths are:
- The proposed method has a nice connection with Adam and SGD, which helps further understanding of optimizations algorithms. This paper provided discussions on various approaches for search directions (one-dimension or full-dimension) and convergence theory for DRAG using standard assumptions.
- Experiment results show encouraging performance compared to prior methods.


The weaknesses are
- The subproblem is not motivated well enough. It is hard to find an explanation why the exact objective function of the subproblem should be minimized. Other details may explain that the algorithm are searching for the direction in a subspace of 2-dimension, but it does not explain their choice of objective function. The authors says "update is optimal with respect to the quadratic approximation of the loss function", however solving for that subproblem does not necessary guarantee a good descent direction.
- Some details related to the algorithm and convergence theory is unclear, thus it is hard for the readers to fully understand this method. (Please see the comment below.)

Comments:
- It is unclear when the authors state "We omit $\epsilon$ here and below for discussion simplicity.", but mentioning $\epsilon$ in the complexity later in the paragraph (Page 4). In my understanding, this epsilon maybe the accuracy in approximating the subproblem. If that is true, this complexity should be included in the consideration of total algorithm complexity. In addition, it is not clear if the algorithm use an exact solution of the subproblem or an approximation. Please consider this in the convergence results as well.
- From the context of theorems I assume $\alpha_{1t}$ and $\alpha_{2t}$ should be non-negative. However, I cannot find any justification/explanation on this. Hence the questions:
1. If true, can you provide a theoretical reason why $\alpha_{1t}$ and $\alpha_{2t}$ should be non-negative?
2. The theorems assume that those solutions are lower bounded. Why this may hold true in practice? (if the assumption does not hold, there is no theoretical guarantee for this method.)

- It is not really fair to say that the theory has the "same assumptions" as other methods because the assumptions on $\alpha_{1t}$ and $\alpha_{2t}$, $c\eta$ and $C$ depends on the particular algorithm.
- The abstract is too long. Please shorten it and put the motivation/explanation to the introduction.

**Summary Of The Paper:**

This paper proposes Dimension-Reduced Adaptive Gradient Method (DRAG), a combination of SGD and Adam by solving a trust-region sub problem. The method is inspired by DRSOM (Zhang et al., 2022) which uses Hessian-vector products in two directions. The authors show convergence theory and experiment results for DRAG.

**Summary Of The Review:**

This paper has some unclear points regarding the subproblem in the algorithm and the convergence analysis. I will be more open to accept this paper if the authors clarify these points.

---

> ### Author Response · Authors · 2022-11-14
> **Part 1 of Response to Reviewer Ymav**
>
> We thank the reviewer for the insightful and  encouraging comments, and also very good suggestions. Our point-by-point responses to the questions are given below, and we also look forward to the subsequent discussion that may further help to resolve the current issues.
>
> (1) For the **subproblem of DRAG**, we are motivated by Adam. Adam adopts the update schedule $x_t = x_{t-1} - \eta \frac{\hat{v} _ t}{\sqrt{\hat{u} _ t} + \nu}$. Its update direction is optimal considering the quadratic approximation of $f$ at $x_{t-1}$ in that
> $
> -\frac{\hat{v} _ t}{\sqrt{\hat{u} _ t} + \nu} \approx \operatorname{argmin}_{p} \langle \nabla f(x _ {t-1}), p \rangle +\frac{1}{2}\langle p,H_f(x _ {t-1}) p \rangle
> $
> with the gradient $\nabla f(x _ {t-1})$ approximated by $\hat{v} _ t$ and the Hessian $H_f(x _ {t-1})$ approximated by $\operatorname{diag}(\sqrt{\hat{u} _ t} + \nu)$.
>
> Inspired by the above idea, DRAG adopts the same quadratic approximation  but restricts $p=\alpha _ {1t} \hat{v} _ t + \alpha _ {2t} d_t $ with  an additional trust-region constraint $\|p\| \le \eta \|\hat{v} _ t\|$  to optimize the loss along two fixed directions $\hat{v} _ t$ and $d_t $.  In this way, one can calculate two locally optimal stepsizes $\alpha _ {1t}$ and $\alpha _ {2t}$ for the two directions to minimize the loss. Here we follow Adam's approximation,  since  i) Adam's gradient and Hessian approximation are effective for deep learning, making Adam and its variants the most popular optimizer for training neural networks, ii) the original Hessian $H_f({x _ {t-1}})$,  a $n\times n$ matrix, is computational prohibitive in neural networks. Indeed, DRAG's convergence analysis and good practical performance  justify the trust-region framework.
>
> (2) **For $\epsilon$**, we apologize for using $\epsilon$ many times with different meanings in our paper which could mislead the readers. For clarity, we **replace the small positive constant $\epsilon$ in the Hessian approximation**, which is used for improving the stability of algorithms, with a new notation $\nu$. In this way, remaining $\epsilon$ only denotes the optimization accuracy.
>
> The omission of the small positive constant $\nu$ for the formulation of the two-dimensional trust-region subproblem is just for the simplicity of notations, we now use $H_t$ to further simplify the formulation in the paper.
>
> Regarding **$\epsilon$ in the complexity of solving the subproblem** (page 4), we quote the result in [1]: an $\epsilon$-global minimizer of a trust-region subproblem can be computed in complexity  $\mathcal{O}(\log \log (\frac{1}{\epsilon}))$. In practice, we run no more than 20 iterations to get the solution of the trust-region subproblem. For the **overall complexity**, it does not depend on the complexity of solving the subproblem. This is because 1) the overall complexity denotes the number of stochastic gradient evaluation, and thus specifically refers to gradient complexity; 2) the subproblem is two-dimensional, and therefore its computational cost is marginal compared with the evaluation of stochastic gradient, especially for deep networks.
>
> (3) For the **assumption on $\alpha_{1t}$ and $\alpha_{2t}$ in our convergence analysis**, we don't require $\alpha_{2t}$ to be non-negative lower bounded. The assumption actually contains two part: i) $|\frac{\alpha_{1t}}{\eta_t}|,|\frac{\alpha_{2t}}{\eta_t}|$ are upper bounded by a constant $G$; ii) $\alpha_{1t}$ is positive and $\frac{\alpha_{1t}}{\eta_t}$ is lower bounded by a constant $\delta$. We clarify the reasons behind these assumptions in Appendix B in the revision of the paper.
>
> The main idea is that for i), with the widely used assumption that stochastic gradient per  iteration is bounded, the trust-region constraint ensures that both $|\frac{\alpha_{1t}}{\eta_t}|$ and $|\frac{\alpha_{2t}}{\eta_t}|$ are upper bounded by a constant $G$. For ii), it is based on intuition and observations from experiments. The intuition is that  since the gradient direction is the most important direction and the most reliable information we can obtain at each iteration,  we should at least adopts a positive stepsize for the gradient descent direction. For experiments, our empirical results show that $\frac{\alpha_{1t}}{\eta_t}$ is always larger than 0.1 along the whole training process under all the experimental settings. So we assume $\frac{\alpha_{1t}}{\eta_t} \ge \delta > 0$ in our convergence analysis.

---

> > ### Author Response · Authors · 2022-11-14
> > **Part 2 of Response to Reviewer Ymav**
> >
> > (4) The  **"same assumptions"** means that we impose the same assumptions on the problem and the stochastic gradient, e.g.  i) Lipschitz smooth non-convex loss objective $f$, ii) $f$ is only accessible through an unbiased gradient estimator $\tilde{\nabla} f$, iii) bounded  stochastic gradient and bounded stochastic gradient variance.  These assumptions together induce the stochastic gradient complexity lower bound for non-convex stochastic optimization in [2].  The assumption on the stepsizes $\alpha _ {1t}, \alpha _ {2t}$ does not affect the complexity lower bound, since it does not affect the problem's properties. We have clarified this part in the revision.
> >
> > (5) In the end, we also appreciate the reviewer's advice on the length of our abstract. We have deleted some detailed descriptions of Adam and shortened our abstract.
> >
> > [1] David G Luenberger, Yinyu Ye, et al. Linear and nonlinear programming, volume 2. Springer, 1984.
> >
> > [2] Yossi Arjevani, Yair Carmon, John C Duchi, Dylan J Foster, Nathan Srebro, and Blake Woodworth.
> > Lower bounds for non-convex stochastic optimization. Mathematical Programming, pp. 1–50,
> > 2022.

---

### Official Review · Reviewer_Guez · 2022-10-23

**Confidence:** 4
**Clarity, Quality, Novelty And Reproducibility:** The work is mostly clear.
**Correctness:** 3
**Technical Novelty And Significance:** 2
**Empirical Novelty And Significance:** 2
**Recommendation:** 5

**Strength And Weaknesses:**

Strength:
  1. The proposed algorithm is easy to implement and understand.
  2. The paper is well-written and easy-to-follow.
  3. The experimental results show advantages over traditional methods.

Weakness:
  1. This algorithm seems to be just another variant of Adam in the vast number of similar works, where too many algorithms have been proposed and all claim to be better than Adam, with no theoretical gain and only experimental better performance on some datasets. The contribution of this paper seems insufficient to me.

  2. My first question is, why is the algorithm designed in this particular way? I can kind of understand why and how the trust-region problem is formulated, but why is $H_t$ in line 7 of Algo 1 chosen to be the denominator of Adam? Is there a particular reason why the authors choose Adam? What if I replace it by the simple average of gradient squares (variant of AdaGrad) instead of the exponential moving average? In line 10 of Algo 1 and Sec 3.2, the update seems to be a linear combination of the past gradients $g_1, g_2, ..., g_t$, and $H_t$ is not involved expect for the computation of $\alpha_1$ and $\alpha_2$. If the authors only use $H_t$ to be an estimate of, say the Hessian matrix, then I think any similar estimate can be such matrix? Please correct me if I am wrong.

  3. The theoretical results (Sec 4) do not show any advantage over SGD or Adam. It only proves that the algorithm converges in the nonconvex setting. Although this is appreciated, it does not show that the proposed algorithm can, at least have similar property as Adam in the convex case, i.e., in the convex problem, when the gradients are small or sparse, adaptive algorithms such as AdaGrad and Adam can converge faster than SGD, which is a major reason why people like John Duchi have proposed adaptive algorithms in the first place [1].

  4. The experimental results look quite promising. However, they are all on small datasets with small neural networks. I would be much more convinced if the authors can provide results such as, on ImageNet with ResNet50, on One Billion Word with LSTMs, or similar large-scale experiments. Other kinds of experiments such as Image Generation (GAN), Image Segmentation would also be helpful.


[1]. John Duchi, Elad Hazan, and Yoram Singer. Adaptive subgradient methods for online learning and stochastic optimization. Journal of Machine Learning Research, 12(7), 2011.

**Summary Of The Paper:**

This paper proposes yet another variant of the Adam optimizer, by combining the idea of trust-region problem with adaptive optimization method. The authors provide some heuristics of creating this method and some theoretical analysis of the algorithm. Experimental results validate their claim that DRAG performs better than the other adaptive algorithms.

**Summary Of The Review:**

In summary, the paper is well-written and easy-to-understand. However, I don't think the contributions are enough to make the paper accepted. I would increase my score if the authors can

1. Provide theoretical results showing that DRAG can converge faster than SGD/Adam in convex/nonconvex problems.

2. Provide experimental results on large datasets/other datasets.

---

> ### Author Response · Authors · 2022-11-14
> **Part 1 of Response to Reviewer Guez**
>
> We thank the reviewer for the insightful and  encouraging comments, and also very good suggestions. In the following, we provide our point-by-point responses and hope to address the reviewer's concerns.
>
> (1) Our DRAG is not just a simple variant of the Adam optimizer, it is different from Adam and its variants in two aspect: i) DRAG optimizes the loss along $k$ directions, while Adam and its variants indeed minimizes the loss along $n$ directions, where $n$ is much larger than $k$. The benefits of DRAG is that it can trade off   the $n$-direction search of Adam and the single-direction search direction of SGD, and thus can improve generalization performance. This improvement is intuitively explain in the manuscript and also justified by our experimental results. ii) At each iteration,  DRAG solves a trust-region subproblem to compute the optimal stepsizes for its $k$ directions, while Adam and its variants  use fixed stepsizes which may impair the convergence speed.
>
>
>  For the convergence analysis, we cannot prove the better complexity, since  the complexity of DRAG and other methods actually match the lower bound under the setting of non-convex optimization.  But we want to emphasize that for deep learning community,   a practical optimizer with good generalization performance is highly desired and also very important. So we follow this spirit to design our DRAG and hope it contributes to the community.
>
>
> (2) For the **formulation of the trust-region subproblem**, we are motivated by Adam. Adam adopts the update schedule $x_t = x_{t-1} - \eta \frac{\hat{v} _ t}{\sqrt{\hat{u} _ t} + \nu}$, its update direction is optimal considering the quadratic approximation of $f$ at $x_{t-1}$ in that
> $
> -\frac{\hat{v} _ t}{\sqrt{\hat{u} _ t} + \nu} \approx \operatorname{argmin}_{p} \langle \nabla f(x _ {t-1}), p \rangle +\frac{1}{2}\langle p,H_f(x _ {t-1}) p \rangle
> $
> with the gradient $\nabla f(x _ {t-1})$ approximated by $\hat{v} _ t$ and the Hessian $H _ f(x _ {t-1})$ approximated by $\operatorname{diag}(\sqrt{\hat{u}_t} + \nu)$.
>
> Inspired by the above idea, DRAG adopts the same quadratic approximation  but restricts $p=\alpha _ {1t} \hat{v} _ t + \alpha _ {2t} d_t $ with  an additional trust-region constraint $\|p\| \le \eta \|\hat{v} _ t\|$  to optimize the loss along two fixed directions $\hat{v} _ t$ and $d_t $.  In this way, one can calculate two locally optimal stepsizes $\alpha_{1t}$ and $\alpha_{2t}$ for the two directions to minimize the loss. Here we follow Adam's approximation,  since  i) Adam's gradient and Hessian approximation are effective for deep learning, making Adam and its variants the most popular optimizer for training neural networks, ii) the original Hessian $H_f({x _ {t-1}})$,  a $n\times n$ matrix, is computational prohibitive in neural networks. Indeed, DRAG's convergence analysis and good practical performance  justify the trust-region framework.
>
>
> We agree with you that the **Hessian approximation $H_t$** in the trust-region subproblem is not restricted to Adam's choice. Indeed,  this is also one benefit of our trust-region framework, since our choice of Hessian approximation is more flexible. In practice, as long as the quadratic term $\frac{1}{2}\langle p, H_f(x_{t-1}) p\rangle$ can be computed efficiently, any choice of Hessian approximation is acceptable for our framework.
>
> While we also explored several Hessian approximations, e.g.  positive diagonal matrix, indefinite diagonal matrix, and diagonal plus rank-one correction matrix (quasi-newton type), Adam's approximation performs best among our choices. This is the main reason why we chose to use Adam's approximation. We thank the reviewer for providing another choice of Hessian approximation. We use it to train ResNet34 on CIFAR10 dataset,  and get a test accuracy of 95.2\%, which is lower than the one used in DRAG. Note that although we have made  exploration  to the Hessian approximation, this does not  mean that Adam's choice is the best. We welcome further works that aim to find better Hessian approximations.

---

> > ### Author Response · Authors · 2022-11-14
> > **Part 2 of Response to Reviewer Guez**
> >
> > (3) For providing **theoretical analysis** under the setting of online learning, we appreciate this advice. However, it is impossible for us to provide a rigorous proof within a short period of time. Besides, our algorithm DRAG is mainly designed for network training which is a non-convex optimization problem. Therefore, our convergence analysis targets at the non-convex setting. We will continue to work on the proof under a convex setting and see if our algorithm also has theoretical benefits for convex optimization.
> >
> > (4) For **large-scale experiments**, we have tested ResNet18 on ImageNet, and report the  top-1 test accuracy (\%)  below:
> >
> > |**DRAG** | SGD | Adam | AdamW | AdaBound | Padam | AdaBelief |
> > |:----------:|:------:|:-------:|:---------:|:------------:|:--------:|:-----------:|
> > |**70.41** | 70.23 | 63.79 | 67.93 | 68.13 | 70.07 | 70.08 |
> >
> > Here all the results except DRAG's are the best result we found in the literature. This result is updated in the  revision.

---

> > > ### Comment · Reviewer_Guez · 2022-11-19
> > > **Thank you for the rebuttal**
> > >
> > > I thank the authors for providing a nice rebuttal. However, I still feel that this paper needs to be further improved. Therefore, I have kept my rating.

---

### Official Review · Reviewer_kEoQ · 2022-10-25

**Confidence:** 4
**Clarity, Quality, Novelty And Reproducibility:** see main comments
**Correctness:** 3
**Technical Novelty And Significance:** 2
**Empirical Novelty And Significance:** 2
**Recommendation:** 5

**Strength And Weaknesses:**


Strength:
1. a new adaptive gradient method for better generalization performances
2. no extra hyper-parameters

Weakness
1. improvement is not significant at all
2. does not have very clear intuitions on why the strategy would help
3. theoretical result does not reflect the improvement


**Summary Of The Paper:**

In this paper, the authors proposed a Dimension-Reduced Adaptive Gradient Method (DRAG) to eliminate the generalization gap of adaptive gradient method. DRAG makes a combination of SGD and Adam by adopting a trust-region like framework. DRAG is compatible with the common deep learning training pipeline without introducing extra hyper-parameters and with negligible extra computation.

**Summary Of The Review:**

1. The intuition for the proposed methods is not quite convincing. Why the two-dimensional search is better than a one-dimension or d-dimensional case? Seems a bit unnatural

2. There are many approximations/vague parts in the process, e.g., the Hessian is approximated, and the trust-region is set without many justifications, why would such a trust-region solution give better estimates on the learning rate?

3. Continue with point 2, I am concerned that the optimal lr may only give you better convergence rather than generalization. For example, running vanilla SGD/Adam with line searched lr may not give you better results. I would suggest the authors to compare the derived best lr at the beginning of Page 5 vs the vanilla SGD and the line searched SGD, as well as the Adam version.

4. There are no major improvements shown in experiments. Currently the advantage of DRAG is quite marginal. Also the theoretical result only gives a convergence rate matching existing solutions, which does not give any new information on the generalization.

5. The following method also attempts to combine SGD with Adam for better generalization. The authors may want to comment on/compare with it since the goal is the same.

[1] "Closing the generalization gap of adaptive gradient methods in training deep neural networks." IJCAI2020

---

> ### Author Response · Authors · 2022-11-14
> **Part 1 of Response to Reviewer kEoQ**
>
> We thank the reviewer for the insightful and encouraging comments, and also very good suggestions. Our point-by-point responses to the reviewer's questions are as follow.
>
> (1) For the  **intuition of DRAG**, it is to use a trust-region-like framework to **optimize the network  parameters along $k$ ($\ll n$) directions**, where $k=2$ in the manuscript. This optimization along the $k$ directions is a trade-off between the optimization of  $n$-direction  (full space) in  Adam and the optimization of a single direction in SGD. Compared with SGD, DRAG inherits the optimization along the  gradient  direction in SGD, but also update the parameter along an extra momentum direction, which boosts the convergence speed of DRAG in practice. Moreover, our  trust-region-like framework computes the optimal stepsizes for the two descent directions in DRAG which can further improve the convergence speed.  Unlike Adam, DRAG  largely  reduces the flexibility of update direction from searching in the whole parameter space to updating along the gradient and momentum direction. In this way,  DRAG is less susceptible to overfitting which causes the inferior generalization performance of Adam,  and we could expect better generalization performance of DRAG in practice. Overall, DRAG targets to pursuit the merits of both SGD and Adam via making a trade-off between their optimization directions.
>
> (2) For the **trust-region subproblem of DRAG**, we are motivated by Adam. Adam adopts the update schedule $x_t = x _ {t-1} - \eta \frac{\hat{v} _ t}{\sqrt{\hat{u} _ t} + \nu}$, its update direction is optimal considering the quadratic approximation of $f$ at $x _ {t-1}$ in that
> $
> -\frac{\hat{v} _ t}{\sqrt{\hat{u} _ t} + \nu} \approx \operatorname{argmin}_{p} \langle \nabla f(x _ {t-1}), p \rangle +\frac{1}{2}\langle p,H_f(x _ {t-1}) p \rangle
> $
> with the gradient $\nabla f(x _ {t-1})$ approximated by $\hat{v} _ t$ and the Hessian $H_f(x _ {t-1})$ approximated by $\operatorname{diag}(\sqrt{\hat{u} _ t} + \nu)$.
>
> Inspired by the above idea, DRAG adopts the same quadratic approximation  but restricts $p=\alpha_{1t} \hat{v} _ t + \alpha _ {2t} d_t $ with  an additional trust-region constraint $\|p\| \le \eta \|\hat{v} _ t\|$  to optimize the loss along two fixed directions $\hat{v}_t$ and $d_t $.  In this way, one can calculate two locally optimal stepsizes $\alpha _ {1t}$ and $\alpha _ {2t}$ for the two directions to minimize the loss. Here we follow Adam's approximation,  since  i) Adam's gradient and Hessian approximation are effective for deep learning, making Adam and its variants the most popular optimizer for training neural networks, ii) the original Hessian $H_f({x _ {t-1}})$,  a $n\times n$ matrix, is computational prohibitive in neural networks. Indeed, DRAG's convergence analysis and good practical performance  justify the trust-region framework.
>
> (3) For **better generalization performance of DRAG over SGD**, we attribute it to two factors:  **i) an extra momentum search direction in DRAG**; and **ii) optimal stepsizes in DRAG.**
>
> For i), DRAG updates the variable $x$ in two directions at each iteration, while SGD only updates along a single direction. Therefore, DRAG is more likely to find better local optima.
>
> For ii), optimal stepsize makes the algorithm converge faster, and thus allow the algorithm to approach a local minimum closer with given number of training iterations. Moreover, the optimal stepsize can relieve oscillation of the algorithm at the end of training, especially when the training parameters are already close to a local minima.
>
>
> As per your suggestion,  we had wanted to compare an algorithm with its **line search version**. However, for  the real  network training,   line search  is not practical because of the prohibitive computational overhead of evaluating the loss function several times in a single training iteration to find the best stepsize. On the other hand, we extend our trust region framework to SGD so that at each iteration, we can find a locally optimal stepsize for SGD. On ResNet34 with CIFAR10, SGD with trust region stepsize achieves  95.3\%  test accuracy which is actually higher than the accuracy of   94.5\% obtained by vanilla SGD. This experimental result  can well support ii). Moreover,  our DARG obtains an accuracy of 95.6\%, which is higher than 95.3\%  for SGD with trust-region stepsizes. This result can support i), since the only difference between these two algorithms is  that DARG uses two directions (gradient direction and momentum direction) to optimize the loss while SGD only adopts one gradient direction.

---

> > ### Author Response · Authors · 2022-11-14
> > **Part 2 of Response to Reviewer kEoQ**
> >
> > (4) The **experimental improvement of DRAG**  is not marginal if we compare it with Adam and SGD. Even when compared with delicate variants of Adam such as AdaBelief, DRAG still outperforms it by a large margin under some settings. For example, on  DenseNet121 with CIFAR100, DRAG attains an accuracy of 79.2\%, while the second best method,  AdaBelief,  only attains 78.2\%. The gap between DRAG and AdaBelief is 1\%, which is two times greater than the gap between AdaBelief to SGD (0.4\%).
> >
> > For the convergence analysis, we prove that DRAG has the stochastic gradient complexity matching the lower bound under the setting of non-convex optimization, which helps to explain DRAG's fast convergence behavior in practice.
> >
> > For generalization, the fact the DRAG's update lies in the subspace $\operatorname{span} ( g_0, g_1, \cdots, g_{t-1}  ) $  guarantees that at least for binary classification problem, DRAG always converges to the max-margin solution while adaptive gradient methods generalize poorly. This helps to explain DRAG's generalization advantage over adaptive gradient methods.
> >
> > (5) We appreciate the reviewer's suggestion on comparing our method with Padam. In the revision, we have added Padam to related work  and the experiments.

---

### Official Review · Reviewer_LdvC · 2022-10-25

**Confidence:** 4
**Correctness:** 3
**Technical Novelty And Significance:** 3
**Empirical Novelty And Significance:** 3
**Recommendation:** 5

**Clarity, Quality, Novelty And Reproducibility:**

Quality:
Current quality is fine but not extremely good. (Maybe I misunderstood Thm1 and Lemma4, wait for the authors' response).

Clarity:
good

Originality:
good

**Strength And Weaknesses:**

Strength:
1) The perspective of update freedom for comparing Adam vs SGD is novel to me. Furthermore, the authors formalize the trust-region idea, and limits the update direction to be a linear combination of first-order momentum and Nesterov-update, this limits the update at each step to be in the convex hull of all past gradients, which was shown to not hurt generalization performance as Adam.
2) The authors conducted experiments on various architectures (CNN and LSTM) and compare with most of recently proposed optimizers, which is a more thorough comparison.

Weakness:
1) I'm not fully convinced by Thm1. Specifically, I checked the proof which should be correct up to Lemma 4. When deriving Thm1 from Lemma4, it does not look fully correct to me, because in Lemma4 RHS there's a term $C_2 (1-\beta) \sigma^2$, which does not goes to 0 as $T \to \infty$. This conclusion does not show that the risk $\frac{1}{T} \sum \mathbb{E} ||\nabla f(x_t)||^2 \to 0 as T \to \infty$, which means Lemma4 does not fully prove the convergence of the algorithm.

For proof of Thm1, the condition that $C_2 (1-\beta) \sigma^2 \leq \epsilon^2 / 3$ is again pretty tricky, because $\sigma^2$ is the noise of gradient which you can not control, $C_2$ is another constant that depends on $\eta$, $C_2 = \frac{4G}{\delta} \geq \frac{\eta}{\alpha}$ without an upper bound, (I derive this based on Lemma4's assumption $0<\delta<\frac{\alpha}{\eta}$, which means this term in quite hard to bound unless you make an extra assumption on the lower-bound of $\eta$.

(I might be missing some point here, if the authors could correct me I would be happy to increase rating)

2) The experiments are OK but not very complete. I would suggest the authors to test the proposed method on tasks where Adam typically outperforms SGD, for example transformers and reinforcement learning. For current experiments SGD almost always outperforms Adam.

**Summary Of The Paper:**

The authors observe that Adam has too much freedom in optimization, while SGD has one direction (too little freedom), hence the authors propose to propose an algorithm that has flexibility between SGD and Adam to achieve the benefits of both. Specifically, the authors proposed DRAG, which combines Adam with a trust-region like framework, and optimize the loss along $k$ descent directions, where $k$ is a pre-defined hyper-parameter ranging between 1 and $n$ ($n$ is the number of parameters).

The authors also proved the convergence of DRAG in the stochastic non-convex optimization setting, and validated DRAG with experiments for CNN and LSTM.

**Summary Of The Review:**

The authors propose an interesting and novel idea, which controls the freedom of update to be between Adam and SGD. The analysis is in general complete and experiments are satisfactory. I'm concerned that Thm1 might be wrong, but honestly I think Thm1 is unnecessary and just Thm2 is sufficient. I'll adjust my rating according to the authors' response.

---

> ### Author Response · Authors · 2022-11-14
> **Response to Reviewer LdvC**
>
> We thank the reviewer for the insightful and encouraging comments, and also very careful reading!  In the following,  we provide our response to the questions and hope to address the reviewer's concerns as much as possible. We also look forward to the subsequent discussion which may further help to resolve the current issues.
>
> We check our proof again, and the conclusion is that our proof has no problem. However, for the convergence analysis of DRAG, we do apologize for not clarifying the assumptions, resulting in difficulties for the readers to understand the theorems. Except for Assumption 1 in the paper, we have two additional assumptions that make our analysis work.
>
> First, for Theorem 1 and Theorem 2, we have the assumption that **the first-order momentum parameter $\beta_1$ is approximately equal to 1 (increasingly)**. This assumption is different from some previous analyses, such as [1], which uses a first-order momentum parameter that decreases to zero. From our knowledge, [2] is the first to use this assumption to prove the convergence of vanilla Adam, which is proved to diverge in [1] under its assumption of a decreasing momentum parameter $\beta_1$. The proof of [2] helps to explain the practical success of Adam. Moreover, a large first-order momentum parameter $\beta_1$ is what we use in practice, and [2] experimentally shows that a stage-wise increasing schedule of $\beta_1$ improves the performance of Adam. Therefore, we follow the assumption of [2], and assume a first-order momentum parameter $\beta_1$ that increasingly approximates 1.
>
> For the derivation in Lemma 4, we can prove the result, $C _ 2(1-\beta)\sigma^2 \leq \frac{\epsilon^2}{3}$, because in Theorem 1,  we assume  $\beta$ satisfies $1-\beta \le \frac{\epsilon^2}{3C_2\sigma^2}$.  Similarly,   in Theorem 2 we assume  $\beta_t$ to obey  $1-\beta_t = \frac{C c_\eta}{\sqrt{t+1}}$ and thus we have the conclusion that   $C_2(1-\beta)\sigma^2$ goes to 0 as $T \rightarrow \infty$.
>
> Second, we also have an **assumption on the stepsizes $\alpha_{1t},\alpha_{2t}$**, which includes two parts: **i) $|\frac{\alpha_{1t}}{\eta_t}|,|\frac{\alpha_{2t}}{\eta_t}|$ are upper bounded by a constant $G$, ii) $\alpha_{1t}$ is positive and $\frac{\alpha_{1t}}{\eta_t}$ is lower bounded by a positive constant $\delta$. Details on why the assumption on $\alpha_{1t},\alpha_{2t}$ are mild can be found in Appendix B of the revised paper.** The main idea is that for i), with the widely used assumption that stochastic gradient at each iteration is bounded, the trust-region constraint ensures that both $|\frac{\alpha_{1t}}{\eta_t}|$ and $|\frac{\alpha_{2t}}{\eta_t}|$ are upper bounded by a constant $G$. For ii), it is based on intuition and observations from experiments. The intuition is that  since the gradient direction is the most important direction and the most reliable information we can obtain at each iteration,  we should at least adopts a positive stepsize for the gradient descent direction. For experiments, our empirical results show that $\frac{\alpha_{1t}}{\eta_t}$ is always larger than 0.1 along the whole training process under all the experimental settings. So we assume $\frac{\alpha_{1t}}{\eta_t} \ge \delta > 0$ in our convergence analysis.
>
> With the above assumptions on $\alpha_{1t},\alpha_{2t}$, which is stated as "($\delta,G$) are constants independent with $t$" in Lemma 4, $C_2=\frac{4G}{\delta}$ is a constant. Then we can assume $1-\beta \le \frac{\epsilon^2}{3C_2\sigma^2}$ in Theorem 1.
>
> In the end, we also appreciate the reviewer's suggestion for conducting further experiments under the setting where Adam outperforms SGD. We tried to run Vision Transformer on ImageNet but found that it is impossible for us to finish the experiments in a few days with limited computing resources. Therefore, we choose another classical large-scale setting to further complete our experiments: training ResNet18 on ImageNet. The results of ResNet18 on ImageNet are updated in the revision, and DRAG attained a top-1 test accuracy (\%) of 70.41, outperforming all other algorithms. We will continue to conduct experiments on Vision Transformer to further justify our algorithm.
>
>
> [1] Sashank J Reddi, Satyen Kale, and Sanjiv Kumar. On the convergence of adam and beyond.
>
> [2] Zhishuai Guo, Yi Xu, Wotao Yin, Rong Jin, and Tianbao Yang. A novel convergence analysis for
> algorithms of the adam family.
>
> [3] Juntang Zhuang, Tommy Tang, Yifan Ding, Sekhar C Tatikonda, Nicha Dvornek, Xenophon Papademetris, and James Duncan. Adabelief optimizer: Adapting stepsizes by the belief in observed
> gradients.

---

### Decision · Program_Chairs · 2023-01-20

**Decision:**

Reject

**Justification For Why Not Higher Score:**

Clearly a weak optimization paper.

**Justification For Why Not Lower Score:**

N/A

**Metareview: Summary, Strengths And Weaknesses:**

None of the reviewers judged this paper to be above the acceptance bar. In particular, as one reviewer writes, this paper seems "just another variant of Adam in the vast number of similar works, where too many algorithms have been proposed and all claim to be better than Adam, with no theoretical gain and only experimental better performance on some datasets." Indeed, the theory does not show any advantage over SGD. Note that while the SGD rate is optimal, one could still try to argue for specific cases where this algorithm has advantage or that in practice some terms might have a smaller value than the worst-case ones. Moreover, the experiments are also unconvincing. Overall, the paper is not ready to be published at ICLR.